# TOP-ERL: TRANSFORMER-BASED OFF-POLICY EPISODIC REINFORCEMENT LEARNING

**Ge Li**[*]      **Dong Tian**      **Hongyi Zhou**      **Xinkai Jiang**
**Rudolf Lioutikov**      **Gerhard Neumann**

Karlsruhe Institute of Technology, Germany

## ABSTRACT

This work introduces Transformer-based Off-Policy Episodic Reinforcement Learning (TOP-ERL), a novel algorithm that enables off-policy updates in an ERL framework. In ERL, policies predict entire action trajectories over multiple time steps instead of single per-step actions. These trajectories are typically parameterized by trajectory generators such as Movement Primitives (MP), allowing for smooth and efficient exploration over long horizons while capturing temporal correlations. However, ERL methods are often constrained to on-policy frameworks due to the difficulty of evaluating state-action values for action sequences, limiting their sample efficiency and preventing the use of more efficient off-policy architectures. TOP-ERL addresses this shortcoming by segmenting long action sequences and estimating the state-action values for each segment using a transformer-based critic architecture alongside an n-step return estimation. These contributions result in efficient and stable training that is reflected in the empirical results conducted on sophisticated robot learning environments. TOP-ERL significantly outperforms state-of-the-art RL methods. Thorough ablation studies additionally show the impact of key design choices on the model performance. Our code is available here.

## 1 INTRODUCTION

This work proposes a novel off-policy Reinforcement Learning (RL) algorithm that utilizes a transformer architecture for predicting the values for action sequences. These returns are effectively used to update the policy that predicts a smooth trajectory instead of a single action in each decision step. Predicting a whole trajectory of actions is commonly done in episodic RL (ERL) (Kober & Peters, 2008) and differs conceptually from conventional step-based RL (SRL) methods like SAC (Haarnoja et al., 2018a) where an action is sampled in each time step. The action selection concept in ERL is promising as shown in recent works in RL (Otto et al., 2022; Li et al., 2024). Similar insights have been made in the field of Imitation Learning, where predicting action sequences instead of single actions has led to great success (Zhao et al., 2023; Reuss et al., 2024). Additionally, decision-making in ERL aligns with the human's decision-making strategy, where the human generally does not decide in each single time step but rather performs a whole sequence of actions to complete a task – for instance, swinging an arm to play tennis without overthinking each per-step movement.

**Episodic RL** is a distinct family of RL that emphasizes the maximization of returns over entire episodes, rather than optimizing the intermediate states during environment interactions (Whitley et al., 1993; Igel, 2003; Peters & Schaal, 2008). Unlike SRL, ERL shifts the solution search from per-step actions to a parameterized trajectory space, leveraging techniques like Movement Primitives (MPs) (Schaal, 2006; Paraschos et al., 2013) for generating action sequences. This approach enables a broader exploration horizon (Kober & Peters, 2008), captures temporal and degrees of freedom (DoF) correlations (Li et al., 2024), and ensures smooth transitions between re-planning phases (Otto et al., 2023). Recent advances have integrated ERL with deep learning architectures, demonstrating significant potential in areas such as versatile skill acquisition (Celik et al., 2024) and safe robot reinforcement learning (Kicki et al., 2024). However, despite their advantages, ERL methods often suffer from low update efficiency. Nearly all ERL approaches to date remain constrained to an

---

[*] Accepted as a Spotlight at ICLR 2025. Email to <geli.bruce.ai@gmail.com, ge.li@kit.edu>

on-policy training paradigm, limiting their ability to exploit more efficient off-policy update rules, where an action-value function, or *critic*, is explicitly learned to guide policy updates and action selection. The primary challenge is that prominent off-policy methods, such as SAC (Haarnoja et al., 2018a), rely on temporal difference (TD) error (Sutton, 1988) to update the critic, which implicitly assumes that actions are selected based on each perceived state, rather than a sequence of actions predicted at the start of the episode, as in ERL approaches. In this paper, we address this limitation by predicting the N-step return (Sutton & Barto, 2018) for a sequence of actions using a Transformer architecture, enabling the learning of sequence values within an off-policy framework.

**Transformer in RL.** Over the past few years, the Transformer architecture (Vaswani, 2017) has emerged as one of the most powerful models for sequence data. It has been been integrated into RL across various domains, capitalizing on their strengths in sequence pattern recognition from static datasets and functioning as a memory-based architecture, which aids in task understanding and credit assignment. Applications of Transformers in RL include offline RL (Chebotar et al., 2023; Yamagata et al., 2023; Wu et al., 2024), offline-to-online fine-tuning (Zheng et al., 2022; Ma & Li, 2024; Zhang et al., 2023), handling partially observable states (Parisotto et al., 2020; Ni et al., 2024; Lu et al., 2024), and model-based RL (Lin et al., 2023). However, the use of Transformers within a model-free online RL framework, specifically for sequence action prediction and evaluation, remains largely unexplored (Yuan et al., 2024). This is noteworthy, as similar techniques, such as action chunking (Bharadhwaj et al., 2024), have already proven successful in other domains like imitation learning.

In this paper, we propose **Transformer-based Off-Policy ERL (TOP-ERL)**, which leverages the Transformer as a critic to predict the value of action sequences. Given a trajectory from ERL, we split it into smaller segments and input them into the Transformer for value prediction. We adapt off-policy update rules for action sequences, using the N-step TD error for critic updates. The policy then selects action sequences based on the preferences of the Transformer critic, similar to SAC. Compared to existing ERL and SRL methods, we show that TOP-ERL improves both policy quality and sample efficiency, outperforming them in several simulated robot manipulation tasks. **Our contributions** are: (a) A novel off-policy RL method that integrates the Transformer as a critic for action sequences in a model-free, online RL framework. (b) The use of N-step return as the learning objective for the Transformer critic. (c) Comprehensive evaluation on simulated robotic manipulation tasks, demonstrating superior performance against baselines. (d) Analysis of different critic update rules, design choices, and the impact of segment length on model performance.

## 2   RELATED WORKS

**Episodic RL.** The study of ERL approaches dates back to the 1990s. Early approaches employed black-box optimization techniques to update parameters of policies, such as small MLPs (Whitley et al., 1993; Igel, 2003; Gomez et al., 2008). Due to the substantial data requirements of black-box algorithms and the limited computational resources available at the time, these approaches were constrained to low-dimensional tasks like Pendulum and Cart Pole. Subsequent works (Salimans et al., 2017; Mania et al., 2018) demonstrated that, given sufficient computational resources, ERL methods can also achieve comparable performance to step-based RL on challenge locomotion tasks, such as Ant and Humanoid, at the cost of more samples for convergence. Another line of research in ERL focuses on more compact policy representations. Peters & Schaal (2008) first proposed using movement primitives (MPs) as parameterized policies for ERL, reducing the search space from the high-dimensional neural network parameter space to the MP weight space, which typically ranges from 20 to 50 dimensions, resulting in less samples required for convergence. Using MPs as policies also provides additional benefits, such as smooth trajectory generation and more consistent exploration (Li et al., 2024). MP-based ERL approaches have demonstrated the ability to master complex manipulation tasks such as robot baseball (Peters & Schaal, 2008) and juggling (Ploeger et al., 2021). To further improve sample efficiency, Abdolmaleki et al. (2015) introduced a model-based method to enable more sample-efficient black-box searching. However, these methods are limited in handling tasks with contextual variations, e.g., changing goals. To address this limitation, Abdolmaleki et al. (2017) and Celik et al. (2022) extend MP-based ERL by using linear policies conditioned on context. Otto et al. (2022) enhanced contextual MPRL by employing neural network policies and trust-region regularized policy update. Despite these advances, existing ERL methods generally treat the episodic trajectory as a black box. While this approach allows them to handle sparse and even non-Markovian rewards, ignoring the temporal structure within each episode leads

to lower sample efficiency compared to step-based methods, especially in settings with dense rewards. To address this issue, a most recently proposed method, *Temporally-Correlated ERL* (TCE) (Li et al., 2024) introduced a more efficient update scheme that "opens the black-box" and utilizes sub-segment information for policy update while retaining the benefit of episodic exploration. Although TCE improves the sample efficiency of contextual ERL methods, it still relies on on-policy policy gradient updates, which are considered sample-inefficient. To the best of our knowledge, TOP-ERL is the first off-policy ERL algorithm capable of handling contextual tasks.

**Transformers in model-free RL.** Inspired by the success of Transformers in domains requiring sequence reasoning, the study incorporating Transformers in RL to solve tasks that require long-horizon memory emerged. However, using standard Transformers in RL could results in performance comparable to random policy (Parisotto et al., 2020). To address this issue, *Gated Transformer-XL*(GTrXL) (Parisotto et al., 2020) augmented Transformer-XL with GRU-style gating layers between multi-head self-attention layers, stabilizing the training of deep Transformer networks (up to 12 layers) with online RL. Another research line focuses on utilizing Transformers to enhance offline RL, where the learning process is based on a fixed dataset collected by arbitrary behavior policies. Decision Transformers (Chen et al., 2021) were the first to formulate offline RL as a sequence modeling problem. Subsequent works extended this approach by incorporating dynamic history length adjustment (Wu et al., 2024), Q-learning (Yamagata et al., 2023), and replacing the Transformer with a more efficient state-space model (Ota, 2024). Online Decision Transformers (Zheng et al., 2022) further advanced Decision Transformer by introducing online fine-tuning. In contrast to these studies, which primarily focus on offline RL or fine-tuning pre-trained models, TOP-ERL is designed for online RL and does not rely on offline training. Additionally, TOP-ERL is not designed to solve tasks that require long-horizon memory. Instead, it focuses on using a Transformer-based critic to improve multi-step TD learning within the ERL framework.

## 3 PRELIMINARIES

### 3.1 OFF-POLICY REINFORCEMENT LEARNING

**Markov decision process (MDP).** RL learns policies that maximize cumulative rewards in a given environment, modeled as an MDP. Formally, we consider an MDP defined by a tuple $(\mathcal{S}, \mathcal{A}, P, r, \gamma)$, where both state $\mathcal{S}$ and action spaces $\mathcal{A}$ are continuous. Here, $P(s'|s, a)$ denotes the state transition probability, $r(s, a)$ is the reward function, and $\gamma \in [0, 1]$ is the discount factor. The goal of RL is to find a policy $\pi(a|s)$ that maximizes the expected *return*, which is the sum of discounted future rewards as $G_t(s_t, a_t) = \sum_{i=0}^{\infty} \gamma^i r_{t+i}$.

In **off-policy RL**, the agent learns a policy $\pi(a|s)$ using data generated by a different behavior policy $\pi_b(a|s)$. This enables off-policy methods to reuse past experiences, significantly improving sample efficiency against on-policy methods. A common approach in off-policy RL is to use a *critic*, which estimates the action-value function $Q^\pi(s, a)$ and is updated using a temporal difference (TD) error

$$Q^\pi(s, a) = \mathbb{E}_\pi \left[ G_t \mid s_t = s, a_t = a \right], \quad \delta_t = r_t + \gamma Q^\pi(s_{t+1}, a_{t+1}) - Q^\pi(s_t, a_t), \quad (1)$$

where the TD error $\delta_t$ estimates the difference between the current Q-value and the target Q-value. While the above single-step TD error is useful, it can suffer from high bias and slow convergence, especially in environments with delayed rewards. To address this, N-step returns (Sutton, 1988) are often used to provide a better balance between bias and variance.

**The N-step return** extends the single-step TD return by incorporating multiple future time-steps into the target. Unlike bootstrapping after a single time step, the N-step return accumulates rewards over $N$ steps before using the current value estimate for bootstrapping. These estimates are typically less biased than the 1-step return, but also contain more variance. In off-policy settings, the N-step return typically involves importance sampling (Sutton & Barto, 2018), as the selection of the future action path used to accumulate rewards differs from the current policy $\pi(a|s)$, seen as:

$$G_t^{(N)}(s_t, a_t) = \sum_{i=0}^{N-1} \left( \prod_{j=0}^{i} \rho_{t+j} \right) \gamma^i r_{t+i} + \left( \prod_{j=0}^{N-1} \rho_{t+j} \right) \gamma^N Q^\pi(s_{t+N}, a_{t+N}), \quad (2)$$

where $\rho_t = \frac{\pi(a_t|s_t)}{\pi_b(a_t|s_t)}$ is the importance sampling ratio, ensuring that updates remain unbiased even when using trajectories generated by a different policy.

Despite this mathematical correction, applying N-step returns in off-policy learning can face difficulties, particularly for long sequences. The product of importance ratios can become highly volatile, leading to either exploding or vanishing values over extended trajectories, which in turn can cause high variance in the value estimates and destabilize the learning process. In TOP-ERL, however, we employ N-step return for computing the target value of a sequence of actions, i.e. $G_t^{(N)}(s_t, a_t, a_{t+1}, ..., a_{t+N})$, where N-step actions are determined in a sequence read from the replay buffer, rather than sampled from the policy. Therefore, the resulting formulation does not contain the importance weights. We will further discuss the details in Sec. 4.3.

## 3.2 EPISODIC REINFORCEMENT LEARNING (ERL)

**Episodic RL** (Whitley et al., 1993; Kober & Peters, 2008) focuses on predicting an entire sequence of actions to complete a task, optimizing the cumulative return without explicitly considering detailed state transitions within the episode. Typically, ERL methods utilize a parameterized trajectory generator, such as motion primitives (MP) (Schaal, 2006; Paraschos et al., 2013), which predicts a trajectory parameter vector $\boldsymbol{w}$. This vector is then mapped to a full action trajectory $\boldsymbol{a}(\boldsymbol{w}) = [\boldsymbol{a}_t]_{t=0}$, where $T$ is the trajectory length. Here, $\boldsymbol{a}_t \in \mathbb{R}^D$ denotes the action at time step $t$, and $D$ represents the dimensionality of the action space, such as the degrees of freedom (DOF) in a robotic system. In this framework, an intelligent agent—such as a robot—executes the predicted action sequence directly as motor commands or follows the trajectory using a tracking controller.

Although ERL predicts an entire action trajectory, it still adheres to the *Markov property*, where the state transition probability depends only on the current state and action (Sutton & Barto, 2018). Thus, while the action sequence in ERL spans multiple time steps, the underlying process remains consistent with the MDP formalism. This approach is conceptually related to techniques such as action repeat (Braylan et al., 2015) and temporally correlated action selection (Raffin et al., 2022; Eberhard et al., 2022), which also incorporate temporal dependencies into action selection.

**Movement Primitives (MP)**, as parameterized trajectory generators, play a crucial role in ERL. We briefly highlight key MP methodologies and their mathematical foundations used in this work, with a more detailed discussion in Appendix B. Schaal (2006) introduced Dynamic Movement Primitives (DMPs), which incorporates a forcing term into a dynamical system to generate smooth trajectories from a given initial condition, such as a robot's position and velocity at a particular time[1].

$$\tau^2 \ddot{y} = \alpha(\beta(g-y) - \tau\dot{y}) + f(x), \quad f(x) = x\frac{\sum \varphi_i(x)w_i}{\sum \varphi_i(x)} = x\boldsymbol{\varphi}_x^{\mathsf{T}}\boldsymbol{w}, \tag{3}$$

where $y = y(t)$, $\dot{y} = \mathrm{d}y/\mathrm{d}t$, $\ddot{y} = \mathrm{d}^2y/\mathrm{d}t^2$ denote the position, velocity, and acceleration of the system at time $t$, respectively. Constants $\alpha$ and $\beta$ are spring-damper parameters, with $g$ as the goal attractor and $\tau$ as a time constant modulating the speed of trajectory execution. The functions $\varphi_i(x)$ represents the basis functions for the forcing term, and the trajectory's shape is determined by the weight parameters $w_i \in \boldsymbol{w}$, for $i = 1, ..., N$. The trajectory $[y_t]_{t=0:T}$ is typically computed by numerically integrating the dynamical system from the start to the end. Building on the same concepts, Li et al. (2023) proposed Probabilistic Dynamic Movement Primitives (ProDMPs), which directly uses the closed-form solution of Eq.(3). ProDMP employs a linear basis function representation to directly map a parameter vector $\boldsymbol{w}$ to its corresponding trajectory $[y_t]_{t=0:T}$:

$$y(t) = \boldsymbol{\Phi}(t)^{\mathsf{T}}\boldsymbol{w} + c_1 y_1(t) + c_2 y_2(t). \tag{4}$$

Here, the terms $c_1 y_1(t) + c_2 y_2(t)$ ensure precise trajectory initialization, with the constants $c_1, c_2$ calculated based on the initial condition $y_b, \dot{y}_b$ at time $t_b$. The term $\boldsymbol{\Phi}(t)$ denotes the integral form of the basis functions $\boldsymbol{\varphi}$ used in the Eq.(3). Unlike DMP, ProDMP benefits from the closed-form solution of the dynamic system, enabling faster computation and probabilistic modeling without the burden for numerical integration. This allows for flexible trajectory generation and precise initial condition enforcement. In TOP-ERL, we leverage ProDMP's fast initial condition enforcement to compute accurate target values for the Transformer critic, thereby reducing bias in policy learning.

**ERL Learning Objectives**. A key distinction between ERL and step-based RL (SRL) lies in the action space. ERL shifts the solution search from the per-step action space $\mathcal{A}$ to a parameterized

---

[1]An initial condition in mathematics refers to the value of a function or its derivatives at a starting point, which can be specified at any time and is not necessarily at $t = 0$.

trajectory space $\mathcal{W}$, predicting the trajectory parameters as $\pi(\boldsymbol{w}|\boldsymbol{s})$. As a result, a trajectory parameterized by $\boldsymbol{w}$ is treated as a single data point in $\mathcal{W}$. This often leads ERL to employ black-box optimization methods for trajectory optimization (Salimans et al., 2017). The learning objective in ERL is often formulated using an importance sampling ratio, such as in BBRL (Otto et al., 2022)

$$\text{Update using trajectory parameter:} \quad J = \mathbb{E}_{\pi_{\text{old}}(\boldsymbol{w}|\boldsymbol{s})} \left[ \frac{\pi_{\text{new}}(\boldsymbol{w}|\boldsymbol{s})}{\pi_{\text{old}}(\boldsymbol{w}|\boldsymbol{s})} G^{\pi_{\text{old}}}(\boldsymbol{s}, \boldsymbol{w}) \right], \quad (5)$$

where $\pi$ represents the policy parameterized by $\boldsymbol{\theta}$, typically using a neural network. The terms *new* and *old* refer to the current policy being optimized and the policy used for data collection, respectively. The initial state $\boldsymbol{s} \in \mathcal{S}$ defines the starting configuration and objective of the task, serving as input to the policy. The policy $\pi_{\boldsymbol{\theta}}(\boldsymbol{w}|\boldsymbol{s})$ determines the likelihood of selecting trajectory parameters $\boldsymbol{w}$. The term $G^{\pi_{\text{old}}}(\boldsymbol{s}, \boldsymbol{w}) = \sum_{t=0}^{T} \gamma^t r_t$ represents the return accumulated by executing the trajectory under an old policy, where $\gamma$ is the discount factor and $r_t$ is the reward at time step $t$. By leveraging parameterized trajectory generators like MPs, ERL benefits from consistent exploration, smooth action trajectories, and improved robustness against local optima, as highlighted by Otto et al. (2022). To further enhance learning efficiency, recent work TCE (Li et al., 2024) proposes a hybrid update strategy that decomposes the trajectory parameter-wise update into the segment-wise updates, incorporating per-step information into ERL's learning objective. This approach divides the longer action trajectory into smaller segments, calculating the return of each segment. The new learning objective adapts Eq.(5), with the maximization of segment-wise returns as

$$\text{Update using segments:} \quad J = \mathbb{E}_{\pi_{\text{old}}(\boldsymbol{w}|\boldsymbol{s})} \left[ \frac{1}{K} \sum_{k=1}^{K} \frac{p^{\pi_{\text{new}}}([\boldsymbol{a}_t^k]_{t=0:L}|\boldsymbol{s})}{p^{\pi_{\text{old}}}([\boldsymbol{a}_t^k]_{t=0:L}|\boldsymbol{s})} G^{\pi_{\text{old}}}(\boldsymbol{s}_0^k, [\boldsymbol{a}_t^k]_{t=0:L}) \right], \quad (6)$$

where $K$ and $L$ represent the number and length of the trajectory segments, respectively, with $K = 25$ in the original paper and $k = 1, ..., K$ denotes the segment index. In this expression, $p^\pi$ denotes the likelihood of reproducing the segment, calculated using the parameterized policy $\pi_{\boldsymbol{\theta}}(\boldsymbol{w}|\boldsymbol{s})$, and $G(\boldsymbol{s}_0^k, [\boldsymbol{a}_t^k]_{t=0:L})$ represents the return of executing the $k$-th action sequence segment $[\boldsymbol{a}_t^k]_{t=0:L}$ from the segment's starting state $\boldsymbol{s}_0^k$. It is worth noting, despite the usage of importance sampling, both Eq. (5) and Eq. (6) still remain within the on-policy RL framework. In TOP-ERL, we employ a similar strategy in splitting a long action trajectory into smaller segments, and use these segments for efficient critic and policy updates, under an off-policy framework.

## 4 TRANSFORMER-BASED OFF-POLICY ERL

In this section, we present TOP-ERL, an innovative off-policy solution for ERL that leverages a Transformer for action sequence evaluation. The section is structured as follows: Section 4.1 introduces the Gaussian policy modeling and action trajectory generation, followed by the design of the transformer critic in Section 4.2. The learning objectives for the critic and policy are detailed in Section 4.3 and Section 4.4, respectively, with additional technical details. Lastly, we summarize other design choices in Section 4.5. The main contributions of our model are described from Section 4.2 to Section 4.4, while the remaining sections cover techniques adopted from the literature.

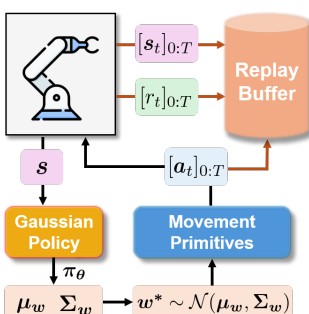

Figure 1: Trajectory generation and environment rollout.

## 4.1 TRAJECTORY GENERATION: TECHNIQUES ADOPTED FROM ERL LITERATURE

TOP-ERL adopts a policy structure similar to previous ERL approaches, such as BBRL (Otto et al., 2022). As shown in Fig. 1, our policy is modeled as a Gaussian distribution, $\pi_{\boldsymbol{\theta}}(\boldsymbol{w}|\boldsymbol{s}) = \mathcal{N}(\boldsymbol{w}|\boldsymbol{\mu_w}, \boldsymbol{\Sigma_w})$, where $\boldsymbol{s}$ defines the initial observation and the task objective, and $\boldsymbol{w}$ represents the parameters of the movement primitive (MPs). In TOP-ERL, we employ ProDMPs (Li et al., 2023) to help correct the target computation via enforcing the initial condition of the MP, as discussed later in Section 4.3.1. Given an initial task state $\boldsymbol{s}$, the policy predicts the Gaussian parameters and samples a parameter vector $\boldsymbol{w}^*$. This vector is then passed into the movement primitive to generate the action trajectory $[\boldsymbol{a}_t]_{t=0:T}$. The agent then executes the action trajectory in the environment until the end of the episode. During the rollout, both the state trajectory and the reward trajectory are recorded. These, along with the action trajectory, are subsequently stored in the replay buffer $\mathcal{B}$ for later use.

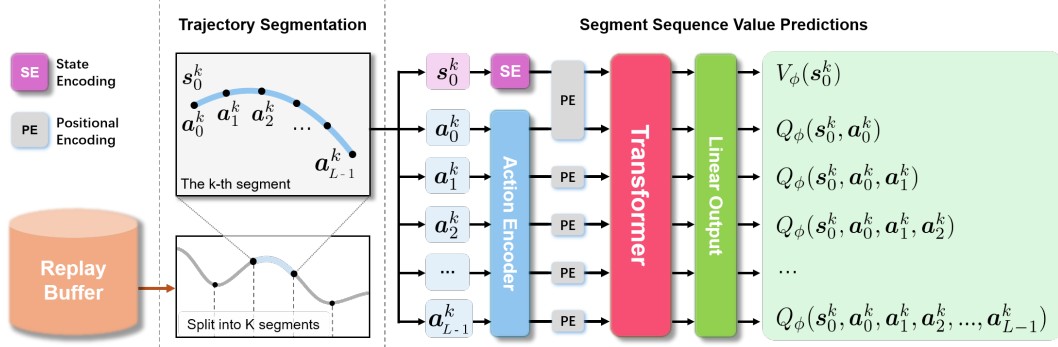

Figure 2: Architecture overview of the Transformer critic, as described in Sec. 4.2.

## 4.2 TRANSFORMERS AS VALUE PREDICTOR FOR ACTION SEQUENCES

An architectural overview of our Transformer critic is depicted in Fig. 2. At each iteration, we sample a batch $B$ of trajectories from the replay buffer and split each trajectory into $K$ segments, where each segment is $L$ time steps long. An ablation on how to select the segment length $L$ can be found in Sec. 5.3. The transformer-based critic has $L + 1$ input tokens that are given by each action in the segment $[a_t^k]_{t=0:L-1}$ and the starting state $s_0^k$ of the corresponding segment. These tokens are first processed by corresponding state and action encoders, each modeled by a single linear layer. Positional information is added to the processed tokens through a trainable positional encoding, with $s_0^k$ and the first action token $a_0^k$ sharing the same positional encoding (both at $t = 0$). The tokens are subsequently fed into a decoder-only Transformer, followed by a linear output layer, producing $L + 1$ output tokens. The first output represents the state value $V(s_0^k)$ for the starting state, while the remaining outputs correspond to the state-action values for the subsequent action sequence. For example, $Q(s_0^k, a_0^k, a_1^k, a_2^k)$ represents the value of executing the actions $a_0^k, a_1^k, a_2^k$ sequentially from the starting state $s_0^k$ and subsequently following policy $\pi$. A causal mask is applied in the Transformer to ensure that actions do not attend to future steps.

## 4.3 N-STEP RETURNS AS THE TARGET FOR TRANSFORMER CRITIC

For each predicted state-action value $Q(s_0, a_0^k, ..., a_{N-1}^k)$ we utilize the N-step return as its target. The objective to update the parameters $\phi$ of the critic is the N-step squared TD error[2]

$$\textbf{Critic loss:} \quad \mathcal{L}(\phi) = \frac{1}{L} \sum_{N=1}^{L-1} \left[ \underbrace{Q_\phi(s_0^k, a_0^k, ..., a_{N-1}^k)}_{\text{Predicted value of N actions}} - \underbrace{G^{(N)}(s_0^k, a_0^k, ..., a_{N-1}^k)}_{\text{Target using \textbf{N-step return}}} \right]^2$$

$$+ \left[ \underbrace{V_\phi(s_0^k)}_{\text{Predicted state value}} - \underbrace{\mathbb{E}_{\tilde{\boldsymbol{w}} \sim \pi_\theta(\cdot|s)} [Q_{\phi_{\text{tar}}}(s_0^k, \tilde{a}_0^k, ..., \tilde{a}_{L-1}^k)]}_{\text{Target of new actions using } \tilde{\boldsymbol{w}}} \right]^2, \quad (7)$$

$$\textbf{N-step return:} \quad G^{(N)}(s_0^k, a_0^k, ..., a_{N-1}^k) = \underbrace{\sum_{i=0}^{N-1} \gamma^i r_i}_{\text{N-step rewards}} + \underbrace{\gamma^N V_{\phi_{\text{tar}}}(s_N)}_{\text{Future return after N-step}}. \quad (8)$$

Here, $N \in [1, L - 1]$ represents the number of actions in a sub-sequence starting from $s_0^k$. The term $Q_{\phi_{\text{tar}}}(s_0^k, \tilde{a}_0^k, ..., \tilde{a}_{L-1}^k)$ in Eq.(7) denotes the target value of $V_\phi(s_0^k)$ with actions $\tilde{a}_0^k, ..., \tilde{a}_{L-1}^k$ generated by new MP parameters $\tilde{w}$ sampled from the current policy, $\tilde{w} \sim \pi_\theta(\cdot \,|s)$. The term $V_{\phi_{\text{tar}}}(s_N)$ in Eq.(8) represents the future return after $N$ steps. Both $Q_{\phi_{\text{tar}}}$ and $V_{\phi_{\text{tar}}}$ are predicted by a target critic (Mnih et al., 2015), with a delayed update rate $\rho = 0.005$. Please note that $Q_{\phi_{\text{tar}}}$ and $V_{\phi_{\text{tar}}}$ are the same transformer network, with and without action tokens.

---

[2]For simplicity, we omit the expectation over buffer $\mathcal{B}$ and average over segment number $K$ in Eq.(7).

In off-policy RL literature, there are several alternatives to replace $V_{\phi_{\text{tar}}}(\boldsymbol{s}_N)$ in Eq.(8). However, we find that this choice alone performs well in our experiments. In other words, TOP-ERL does not necessarily rely on some common off-policy techniques, such as the clipped double-Q (Fujimoto et al., 2018), to be stable and effective. We attribute this to the usage of the N-step returns, which help reduce value estimation bias. In Sec. 5.3, we show that our model can be further improved using these augmentations, though at a cost of additional computation.

Unlike Eq.(2), our N-step return targets $G^{(N)}(\boldsymbol{s}_0^k, \boldsymbol{a}_0^k, \ldots, \boldsymbol{a}_{N-1}^k)$ in Eq.(8) do not include importance sampling as the the action sequence $\boldsymbol{a}_0^k, \ldots, \boldsymbol{a}_{N-1}^k$ is directly used as input tokens for the Q-function. Hence, the actions are fixed and we do not require to compute any expectations over the current policy's action selection. Hence, using the fixed action sequence in Eq.(8) as input to the Q-Function eliminates the need for importance sampling, thus avoiding the high variance typically introduced by it in off-policy methods, as discussed in Sec. 3.1.

### 4.3.1 ENFORCE INITIAL CONDITION FOR NEWLY PREDICTED ACTION SEQUENCE

When calculating the target value $Q_{\phi_{\text{tar}}}(\boldsymbol{s}_0^k, \tilde{\boldsymbol{a}}_0^k, ..., \tilde{\boldsymbol{a}}_{L-1}^k)$ in Eq.(7), a new parameter vector is sampled from the current policy $\tilde{\boldsymbol{w}} \sim \pi_\theta(\,\cdot\,|\boldsymbol{s})$, generating a new action trajectory $[\tilde{\boldsymbol{a}}_t]_{t=0:T}$, with $[\tilde{\boldsymbol{a}}_t^k(\tilde{\boldsymbol{w}})]_{t=0:L-1}$ as a sub-sequence. However, this sequence is not necessarily guaranteed to pass through the segment's starting state $\boldsymbol{s}_0^k$, which creates a mismatch between the state and corresponding action sequence when querying the target in Eq.(7). To address this issue, we append the old reference position to $\boldsymbol{s}_0^k$, and then leverage the dynamic system formulation inherent in ProDMPs by setting the initial condition of the new action sequence to match the old reference at $\boldsymbol{s}_0^k$, as illustrated in Fig. 3. The resulting action sequence $[\tilde{\boldsymbol{a}}_t^k(\tilde{\boldsymbol{w}}, \boldsymbol{s}_0^k)]_{t=0:L-1}$ is therefore depending on both the MP parameters $\tilde{\boldsymbol{w}}_\theta(\boldsymbol{s})$ and the initial condition $\boldsymbol{s}_0^k$. This approach is mathematically equivalent to resetting the initial conditions of an or-

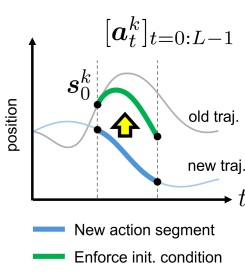

Figure 3: Enforce action initial condition

dinary differential equation (ODE), ensuring consistency between the state and action sequences. Further mathematical details and illustration are provided in Appendix B.3 and B.4.

### 4.4 POLICY UPDATES USING THE TRANSFORMER CRITIC

We utilize the transformer critic to guide the training of our policy, using the reparameterization trick similar to that introduced by SAC (Haarnoja et al., 2018a). The learning objective is to maximize the expected value of the averaged action sequence over varying lengths, defined as:

$$\textbf{Policy Objective:} \quad J(\theta) = \mathbb{E}_{\boldsymbol{s}\sim B}\mathbb{E}_{\tilde{\boldsymbol{w}}\sim\pi_\theta(\cdot|\boldsymbol{s})}\left[\frac{1}{KL}\sum_{k=1}^{K}\sum_{N=0}^{L-1}Q_\phi\left(\boldsymbol{s}_0^k, \left[\tilde{\boldsymbol{a}}_t^k\right]_{t=0:N}\right)\right], \quad (9)$$

where $[\tilde{\boldsymbol{a}}_t^k]_{t=0:N}$ denotes the new action sequence generated by the new MP parameters $\tilde{\boldsymbol{w}}_\theta \sim \pi_\theta(\cdot|\boldsymbol{s})$. This learning objective allows the policy $\pi_\theta(\boldsymbol{w}|\boldsymbol{s})$ to be trained based on the *value preferences* provided by the Transformer critic. We refer Appendix A.1 for more detailed discussion.

### 4.5 ADDITIONAL DESIGN CHOICES FROM THE LITERATURE FOR STABLE LEARNING

We summarize the key learning steps in Algorithm1. To effectively capture a broader range of correlations in both temporal and DoF movements, we utilize a full covariance matrix $\boldsymbol{\Sigma_w}$ in the Gaussian policy (Li et al., 2024). Since the Gaussian policy over MP parameters is typically high-dimensional, we employ the Trust Region Projection Layer (TRPL) (Otto et al., 2021) for stable policy updates, following the design of previous ERL methods (Otto et al., 2022; Li et al., 2024). More discussions are in Appendix A.2. For the Transformer critic, we apply Layer Normalization (Ba, 2016) as the sole data normalization technique, while disabling dropout, as we found it detrimental to performance. In our experiments, we identified the segment length $L$ as a key hyperparameter. The best results were achieved by randomly sampling $L$ at each update iteration, which we attribute to the Transformer critic's ability to attend to different time horizons, resulting in more robust outcomes.

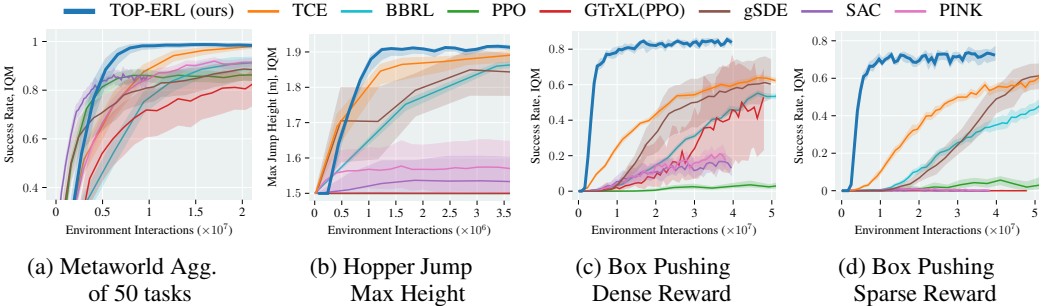

Figure 4: Task Evaluation of (a) Metaworld success rate of 50 tasks aggregation. (b) Hopper Jump Max Height. (c) Box Pushing success rate in dense reward, and (d) sparse reward settings.

## 5 EXPERIMENTS

Our experiments focus on the following questions: I) Can TOP-ERL improve sample efficiency in classical ERL tasks featured by challenging exploration problems? II) How does TOP-ERL perform in large-scale, general manipulation benchmarks? III) How do key design choices affect the performance of TOP-ERL? We compare TOP-ERL against a set of strong baselines. For the ERL comparisons, we select **BBRL** and **TCE** as SoTA ERL methods. For step-based RL, we use **PPO** (Schulman et al., 2017) (on-policy) and **SAC** (off-policy) as established baselines. Additionally, we employed **gSDE** and **PINK**, two step-based RL methods that augment with consistent exploration techniques, to test the impact of exploration strategies.

---

**Algorithm 1** TOP-ERL

1: Initialize critic $\phi$; target critic $\phi_{\text{tar}} \leftarrow \phi$
2: Initialize policy $\theta$ and replay buffer $\mathcal{B}$
3: **repeat**
4:     Reset environment and get initial task state $s$
5:     Predict the policy mean $\boldsymbol{\mu_w}$ and covariance $\boldsymbol{\Sigma_w}$
6:     Sample $\boldsymbol{w}*$ and generate action trajectory $[\boldsymbol{a}]_{0:T}$
7:     Execute the action trajectory till task ends.
8:     Store the visited states $[\boldsymbol{s}]_{0:T}$, rewards $[r]_{0:T}$, and the action $[\boldsymbol{a}]_{0:T}$ trajectories in replay buffer $\mathcal{B}$
9:     **for** each update step **do**
10:         From $\mathcal{B}$, sample a batch of $\boldsymbol{s}, \boldsymbol{a}, r$ trajectories.
11:         Split them into $K$ segments, each $L$ time steps.
12:         Compute N-step return targets as in Eq.(8)
13:         Update transformer critic, using Eq.(7)
14:         Update policy, using Eq.(9)
15:     **end for**
16:     Update target critic $\phi_{\text{tar}} \leftarrow (1 - \rho)\,\phi_{\text{tar}} + \rho\,\phi$
17: **until** converged

---

To assess the impact of using Transformer-based architectures in RL, we include **GTrXL** as baseline for online RL with Transformers architecture. It is worth noting that in the original work, GTrXL was trained using VMPO. However, since the original code was not open-sourced, we used the implementation from Liang et al. (2018), where GTrXL is trained with PPO instead. For all ERLs, the trajectories are generated using ProDMPs with the same hyperparameters and tracked with PD-controllers (or P-controller for MetaWorlds); for all the SRLs, the action outputs are torque (or delta position for MetaWorlds). An overview of the baselines can be find in Table 3, and details regarding the implementation and hyperparameters are provided in the Appendix E.

The evaluation of TOP-ERL are structured in three phases. First, we demonstrated that TOP-ERL significantly improve the sample efficiency over state-of-the-art ERL methods, showcasing its ability to better handle the challenges of sparse rewards and difficult exploration scenarios (Li et al., 2024). Next, we evaluate TOP-ERL on the Meta-World MT50 (Yu et al., 2020) benchmark, a large-scale suite of general manipulation tasks. In this setting, TOP-ERL consistently outperform all baselines, demonstrated TOP-ERL's ability to generalize across a wide range of manipulation tasks. Finally, we conduct a comprehensive ablation study to analysis which ingredient accounts for the strong performance of TOP-ERL. The results confirm that theses components are essential to achieving the strong performance observed with TOP-ERL. To ensure a robust evaluation, all empirical results are reported using Interquartile Mean (IQM), accompanied by a $95\%$ stratified bootstrap confidence interval (Agarwal et al., 2021) across 8 random seeds.

### 5.1 IMPROVING SAMPLE EFFICIENCY IN TASKS WITH CHALLENGING EXPLORATION

ERL methods are renowned for their superior exploration abilities, which often give them an advantage over step-based methods in environments with exploration challenges. However, ERL al-

gorithms are also notoriously sample inefficient, limiting their applicability in scenarios where obtaining samples is expensive. In this evaluation, we investigate whether TOP-ERL can address this limitation by comparing it with baselines on three challenging tasks from Li et al. (2024): Hopper-Jump, a sparse-reward environment where the objective is to maximize the jump height within an episode, and two variants of a contact-rich Box Pushing task. We evaluate the Box Pushing task under both dense and sparse reward settings. Further details about the environments and rewards can be found in Appendix C. The results of these experiments, shown in Fig. 4, demonstrate that TOP-ERL achieved the highest final performance across all three tasks. Notably, in the dense-reward Box Pushing task, TOP-ERL reached an $80\%$ success rate after just 10 million samples, while the second-best method, TCE, only reaches $60\%$ success after 50 million samples. Similar results is observed in the sparse-reward Box Pushing task, where TOP-ERL reaches $70\%$ success rate with 14 million environment interactions, while TCE and gSDE require 50 million samples to reach $60\%$ success. GTrXL performs moderately in the dense-reward setting, achieving a $50\%$ success rate, but fails completely in the sparse-reward environment. Step-based methods like SAC, PINK and PPO failed in both cases, underscoring the difficulty of these tasks. Among the step-based algorithms, only gSDE achieved comparable performance in compare with ERL methods in these three environments, which we attribute to its state-dependent exploration strategy.

## 5.2 CONSISTENT PERFORMANCE IN LARGE-SCALE MANIPULATION BENCHMARKS

In the previous evaluation, we demonstrated that TOP-ERL significantly improves sample efficiency compared to state-of-the-art ERL baselines, while maintaining strong performance in tasks with challenge exploration. In this evaluation, we focus on answering the second question: How does TOP-ERL perform on standard manipulation benchmarks with dense rewards? We conducted experiments on the Meta-World benchmark(Yu et al., 2020), reporting the aggregated success rate **across 50 tasks** in the MT50 task set. To ensure a fair comparison, we followed the same evaluation protocol described in Otto et al. (2022) and Li et al. (2024), where an episode is only considered successful if the success criterion is met at the end of the episode, a more rigours measure than the original setting where success at any time step counts. The results in Fig. 4a show that TOP-ERL achieved highest asymptotic success rate ($98\%$) after 10 million samples. TCE was able to achieve the same success rate but required 20 million interactions. SAC also converged after 10 million samples but with a significantly lower success rate of $85\%$. BBRL and other step-based methods achieved moderate success rate but required significantly more samples.

## 5.3 ABLATION STUDY AND DISCUSSION

**Single Q-Network leads to stable and efficient training.** We compare four common design choices for targets calculation in Q-function update in Eq.(8): 1) V-Target which uses a single V target network, 2) Q-Target, which employs single Q target network, 3)V-Ensemble, which consists of an ensemble of predictions from two V target networks, 4) V-Clip, which takes the minimum of two V target networks. Detailed description of these design choices can be found in Appendix A.3. Fig. 5 presents the learning curves for TOP-ERL in dense-reward (5a) and sparse-reward (5b) Box Pushing, while Table 1 presents the numerical success rate and computation times per update. The results demonstrate that using a single V target network yields performance comparable to approaches that rely on two target networks, with additionally benefit of significantly reduced computation time (approximately 50% faster). We attribute the stable performance with single target network to the use of N-step Bellman equation in target calculation, as discussed in Sec. 4.3.

**Key Components Ablation.** We evaluate the impact of five key components on the performance of TOP-ERL: trust region constraints in policy updates, enforcing the initial condition at each segment, the presence of layer normalization, fixed vs. random segment lengths, and the inclusion of dropout in Transformer layers. These evaluations were conducted in both dense-reward and sparse-reward Box Pushing environments using 8 random seeds. The results, presented in Fig. 5 as dashed lines, show performance for TOP-ERL with corresponding component been added or removed. The results indicate that the random segment length has the most significant effect on TOP-ERL's performance. When using fixed 25 segments the success rate dropped from $80\%$ to $35\%$ in the dense-reward setting, and from $70\%$ to $20\%$ in the sparse-reward setting. Layer normalization, trust region constraints, and enforcing initial conditions also contributed positively to the performance. Interestingly, adding even a small dropout rate (0.05 in the ablation) had negative impacts on the performance in

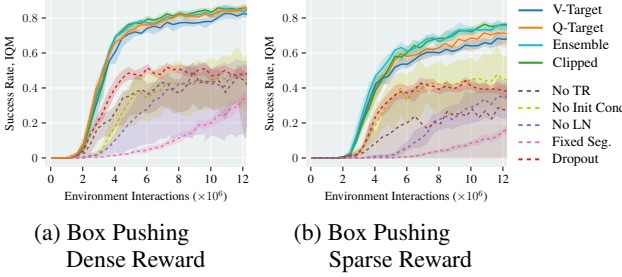

(a) Box Pushing
Dense Reward

(b) Box Pushing
Sparse Reward

Figure 5: Performance of different critic update strategies (solid lines) and model ablations (dashed lines), using Box pushing dense and sparse reward settings respectively.

Table 1: Quantitative performance and update time of different critic update strategies. With additional computational cost, TOP-ERL can be further enhanced.

| Variant | # critic | Time s / iter ↓ | Dense Success, % ↑ | Sparse Success, % ↑ |
|---|---|---|---|---|
| **V-Target** (default) | 1 | **1.55** | 82.0±2.6 | 65.7±4.0 |
| **Q-Target** | 1 | 2.44 | **86.1 ± 2.7** | 69.1 ± 7.5 |
| **V-Ensem.** | 2 | 2.49 | 83.8 ± 3.1 | **75.7 ± 4.4** |
| **V-Clip** | 2 | 2.49 | 86.0 ± 3.2 | 75.5 ± 3.7 |

both tasks. We hypothesize that this effect may be attributed to the use of a relatively small replay buffer combined with a higher buffer update ratio ($0.1\%$ in our setting), which likely mitigates the risk of overfitting in Q-function learning, thereby diminishing the benefit of dropout.

**Impact of Random Segment Lengths.** As shown in the previous ablation study, random segment length played a crucial role in the strong performance of TOP-ERL. To further examine whether this conclusion holds for different segment lengths, we evaluated the dense-reward Box Pushing task with segment lengths ranging from $5\%$ of the episode length to $100\%$ (i.e., no segmentation). The results in Fig. 6 indicate that fixed-length segmentation leads to significant performance variation depending on the segment length. In contrast, random segment lengths consistently achieve faster convergence and higher asymptotic performance. We infer that using a vari-

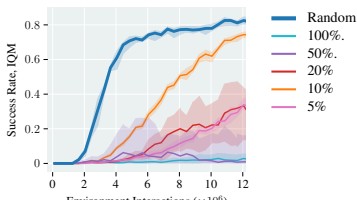

Figure 6: Random or Fixed segment length

ety of action sequence lengths regularizes the training of the critic network. For example, in Eq. (7), the expectation of the Q-value over $L$ actions is used as the target for the V-function's prediction. When $L$ varies across update iterations, the V-function is trained on Q-values derived from different amounts of actions. Additionally, random segmentation simplifies hyperparameter tuning, making it a practical choice. Therefore, we adopt random segmentation length as the default setting for TOP-ERL. To further illustrate the impact of random segment lengths, we provide a visualization of action correlations under different segmentation strategies in Appendix D.

# 6 CONCLUSION

This work introduced Transformer-based Off-Policy Episodic RL (TOP-ERL), a novel off-policy ERL method that leverages Transformers for N-steps return learning. By integrating ERL with an off-policy update scheme, TOP-ERL significantly improves the sample efficiency of ERL methods while retaining their advantages in exploration. The use of a Transformer-based critic architecture allows TOP-ERL to bypass the need for importance sampling in N-steps target calculation, stabilizing training while enjoying the benefit of low-bias value estimation provided by N-steps return. TOP-ERL has demonstrated superior performance compared to state-of-the-art ERL approaches and step-based RL methods augmented with exploration mechanism across 53 challenging tasks, providing strong evidence for its broader applicability to wide range of problems. The ablation studies reveal the reasons behind design choices and components, providing insights into the factors contributing to the strong performance of TOP-ERL.

**Limitations and Future Works.** Despite all the advantages, TOP-ERL shares a limitation common to ERL methods: it generates trajectories only at the start of each episode, making it incapable of handling tasks involving dynamic or target changes within an episode. A promising future research direction would be to incorporate replanning capabilities into TOP-ERL. Additionally, although TOP-ERL uses Transformers as critic, it is not designed to address POMDPs, as the Transformer is used for action-to-go processing in Q-function learning, rather than incorporating state sequences as input. Merging these two paradigms and enhancing TOP-ERL with the ability to handle POMDPs presents another avenue for future investigation.

ACKNOWLEDGMENTS

We thank our friends and colleagues **Juan Li**, **Onur Celik, Aleksandar Taranovic, Tai Hoang and Zuzhao Ye** for their valuable discussion and technical support. We thank the anonymous reviewers for their insightful feedback which greatly improved the quality of this paper.

The research presented in this paper was funded by the Deutsche Forschungsgemeinschaft (DFG, German Research Foundation) – 448648559 and 471687386, and was supported in part by the Helmholtz Association of German Research Centers. Gerhard Neumann was supported in part by Carl Zeiss Foundation through the Project JuBot (Jung Bleiben mit Robotern). The authors acknowledge support by the state of Baden-Württemberg through bwHPC, and the HoreKa supercomputer.

## 7 ETHICS STATEMENT

No human participants were involved in this study. All data used in this work was generated through simulations. As such, there are no privacy or security concerns related to personal or sensitive information. We acknowledge the importance of fairness in AI and have taken care to ensure that our methodology does not introduce bias in simulated environments, though broader fairness issues in real-world applications of such models should be considered in future work. There are no conflicts of interest or sponsorship concerns associated with this research, and all practices adhere to legal and ethical standards.

## 8 REPRODUCIBILITY STATEMENT

The considerable efforts were made to ensure that our work is fully reproducible. All relevant code, including the implementation of the proposed algorithms, simulation environments, and trained models, will be made available in an GitHub repository provided in the main paper. Detailed descriptions of the experimental setup, including hyperparameter configurations can be found in the appendix.

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

# List of Content in Appendix

## A  FURTHER TECHNICAL DETAILS OF TOP-ERL

### A.1  POLICY TRAINING USING SAC-STYLE REPARAMETERIZATION TRICK

We utilize the transformer critic to guide the training of our policy, using the reparameterization trick similar to that introduced by SAC (Haarnoja et al., 2018a). Given a task initial state $\boldsymbol{s}$, the current policy $\pi_{\boldsymbol{\theta}}(\boldsymbol{w}|\boldsymbol{s}) \sim \mathcal{N}(\boldsymbol{w}|\boldsymbol{\mu_w}, \boldsymbol{\Sigma_w})$ predicts the Gaussian parameters of the MP's and samples $\tilde{\boldsymbol{w}}$ as follows:

$$\text{Sample MP parameter vector:} \qquad \tilde{\boldsymbol{w}} = \boldsymbol{\mu_w} + \boldsymbol{L_w}\boldsymbol{\epsilon}, \;\; \boldsymbol{\epsilon} \sim \mathcal{N}(\boldsymbol{0}, \boldsymbol{I}). \tag{10}$$

Here, $\boldsymbol{L_w}$ is the Cholesky decomposition of the covariance matrix $\boldsymbol{\Sigma_w}$, where $\boldsymbol{L_w}\boldsymbol{L_w^T} = \boldsymbol{\Sigma_w}$. This parameterization technique is commonly used for predicting full covariance Gaussian policies. The term $\boldsymbol{\epsilon}$ is a Gaussian white noise vector with the same dimensionality as $\boldsymbol{w}$. Eq. (10) represents the full covariance extension of the reparameterization trick typically used in RL, known as $\tilde{a} = \mu_a + \sigma_a\epsilon, \;\; \epsilon \sim \mathcal{N}(0, 1)$.

The sampled $\tilde{\boldsymbol{w}}$ is then used to compute the new trajectory segments. The resulting trajectory segments are computed using the linear basis function expression in Eq. (4) from Section 3.2, where the coefficients $c_1$ and $c_2$ are determined by the initial conditions and solved using Eq. 22 in Appendix B.3:

$$\text{Compute action segment:} \quad \tilde{\boldsymbol{a}}(t) = \boldsymbol{\Phi}(t)^{\intercal}\tilde{\boldsymbol{w}} + c_1 y_1(t) + c_2 y_2(t) \tag{4}$$

Here, $t = 0 : N$ represents the time interval from the beginning to the end of the $k$-th segment. There are two design choices for the initial conditions: the first is to always use the task initial state $\boldsymbol{s}$ for all segments, while the second is to use the state $\boldsymbol{s}_0^k$ specific to each segment. While the second choice aligns with the techniques described in Section 4.3.1, our empirical results show that the first choice leads to better performance. To the best of our knowledge, we attribute this interesting finding to the following reason. Although the second design choice ensures better consistency in the input to the value function, the per-segment initial conditions were not provided to the policy for action selection, which introduced challenges in updating the policy. We believe this phenomenon is worth further investigation in future research.

We evaluate and maximize the value of these action segments, using their expectation as the policy's learning objective, as shown in Eq. (9) in Section 4.4:

$$\text{SAC style Objective:} \quad J(\theta) = \mathbb{E}_{\boldsymbol{s} \sim B}\mathbb{E}_{\tilde{\boldsymbol{w}} \sim \pi_\theta(\cdot|\boldsymbol{s})}\left[\frac{1}{KL}\sum_{k=1}^{K}\sum_{N=0}^{L-1}Q_\phi(\boldsymbol{s}_0^k, \left[\tilde{\boldsymbol{a}}_t^k\right]_{t=0:N})\right]. \tag{9}$$

Since Eq.(4), (9), (22) and (10) are all differentiable, the policy neural network parameters $\boldsymbol{\theta}$ can be trained using gradient ascent. Compared to the technique introduced in SAC, TOP-ERL adds only one additional step: computing the action sequence $[\tilde{\boldsymbol{a}}_t^k]_{t=0:N}$ from the sampled MP parameter $\tilde{\boldsymbol{w}}$.

## A.2 Utilizing TRPL for Stable Full-Covariance Gaussian Policy Training

In Episodic Reinforcement Learning (ERL), the parameter space $\mathcal{W}$ generally has a higher dimensionality than the action space $\mathcal{A}$, creating distinct challenges for achieving stable policy updates. Trust region methods (Schulman et al., 2015; 2017) are widely regarded as reliable techniques for ensuring convergence and stability in policy gradient algorithms.

Although methods like PPO approximate trust regions using surrogate objectives, they lack the ability to enforce trust regions precisely. To address this limitation, Otto et al. (2021) proposed trust region projection layer (TRPL), a mathematically rigorous and scalable approach for exact trust region enforcement in deep RL algorithms. Leveraging differentiable convex optimization layers (Agrawal et al., 2019), trust region projection layer (TRPL) enforces trust regions at the per-state level and has demonstrated robustness and stability in high-dimensional parameter spaces, as evidenced in methods like BBRL (Otto et al., 2022) and TCE (Li et al., 2024).

TRPL operates on the standard outputs of a Gaussian policy—the mean vector $\boldsymbol{\mu}$ and covariance matrix $\boldsymbol{\Sigma}$—and enforces trust regions through a state-specific projection operation. The adjusted Gaussian policy, represented by $\tilde{\boldsymbol{\mu}}$ and $\tilde{\boldsymbol{\Sigma}}$, serves as the foundation for subsequent computations. The dissimilarity measures for the mean and covariance, denoted as $d_{\mathrm{mean}}$ and $d_{\mathrm{cov}}$ (e.g., KL-divergence), are bounded by thresholds $\epsilon_{\mu}$ and $\epsilon_{\Sigma}$, respectively. The optimization problem for each state $\boldsymbol{s}$ is expressed as:

$$\underset{\tilde{\boldsymbol{\mu}}_s}{\arg\min}\, d_{\mathrm{mean}}\left(\tilde{\boldsymbol{\mu}}_s, \boldsymbol{\mu}(\boldsymbol{s})\right), \quad \text{s.t.} \quad d_{\mathrm{mean}}\left(\tilde{\boldsymbol{\mu}}_s, \boldsymbol{\mu}_{\mathrm{old}}(\boldsymbol{s})\right) \le \epsilon_{\mu}, \text{ and}$$
$$\underset{\tilde{\boldsymbol{\Sigma}}_s}{\arg\min}\, d_{\mathrm{cov}}\left(\tilde{\boldsymbol{\Sigma}}_s, \boldsymbol{\Sigma}(\boldsymbol{s})\right), \quad \text{s.t.} \quad d_{\mathrm{cov}}\left(\tilde{\boldsymbol{\Sigma}}_s, \boldsymbol{\Sigma}_{\mathrm{old}}(\boldsymbol{s})\right) \le \epsilon_{\Sigma}. \tag{11}$$

If the unconstrained, newly predicted per-state Gaussian parameters $\boldsymbol{\mu}(\boldsymbol{s})$ and $\boldsymbol{\Sigma}(\boldsymbol{s})$ exceed the trust region bounds defined by $\epsilon_{\mu}$ and $\epsilon_{\Sigma}$, respectively, TRPL projects them back to the trust region boundary, ensuring stable update steps. In TOP-ERL, the old Gaussian parameters, $\boldsymbol{\mu}_{\mathrm{old}}(\boldsymbol{s})$ and $\boldsymbol{\Sigma}_{\mathrm{old}}(\boldsymbol{s})$, can be derived either from the behavior policy that interacted with the environment or from an exponentially moving averaged (EMA) policy, which serves as a delayed version of the current policy. This approach is analogous to the concept employed in the target critic network.

## A.3 Target Options

Table 2: Options for the future return used in Eq.(8)

| Option | Math | Description |
|---|---|---|
| **V-target** | $V_{\phi}^{\mathrm{tar}}(\boldsymbol{s}_N)$ | State value after N steps |
| **Q-target** | $Q_{\phi}^{\mathrm{tar}}(\boldsymbol{s}_N, \boldsymbol{a}_N, ...)$ | Action value after N steps |
| **Clipped** | $\mathrm{Min}(\cdot, \cdot)$ | Minimum of 2 target critics |
| **Ensemble** | $\mathrm{Avg.}(\cdot, \cdot)$ | Mean of $\ge$ 2 target critics |

## B Mathematical formulations of Movement Primitives.

In this section, we provide an overview of the movement primitive formulations used in this paper. We begin with the basics of DMPs and ProMPs, followed by a detailed explanation of ProDMPs. For clarity, we start with a single DoF system and then expand to multi-DoF systems.

### B.1 Dynamic Movement Primitives

Schaal (2006); Ijspeert et al. (2013) describe a single movement as a trajectory $[y_t]_{t=0:T}$, which is governed by a second-order linear dynamical system with a non-linear forcing function $f$. The mathematical representation is given by

$$\tau^2 \ddot{y} = \alpha(\beta(g - y) - \tau \dot{y}) + f(x), \quad f(x) = x\frac{\sum \varphi_i(x) w_i}{\sum \varphi_i(x)} = x\boldsymbol{\varphi}_x^{\mathsf{T}} \boldsymbol{w}, \tag{12}$$

where $y = y(t)$, $\dot{y} = \mathrm{d}y/\mathrm{d}t$, $\ddot{y} = \mathrm{d}^2y/\mathrm{d}t^2$ denote the position, velocity, and acceleration of the system at a specific time $t$, respectively. Constants $\alpha$ and $\beta$ are spring-damper parameters, $g$ signifies a goal attractor, and $\tau$ is a time constant that modulates the speed of trajectory execution. To ensure convergence towards the goal, DMPs employ a forcing function governed by an exponentially decaying phase variable $x(t) = \exp(-\alpha_x/\tau; t)$. Here, $\varphi_i(x)$ represents the basis functions for the forcing term. The trajectory's shape as it approaches the goal is determined by the weight parameters $w_i \in \boldsymbol{w}$, for $i = 1, ..., N$. The trajectory $[y_t]_{t=0:T}$ is typically computed by numerically integrating the dynamical system from the start to the end point (Pahič et al., 2020; Bahl et al., 2020). However, this numerical process is computationally intensive. For example, to compute the trajectory segment in the end of an episode, DMP must integrate the system from the very beginning till the start of the segment.

## B.2 PROBABILISTIC MOVEMENT PRIMITIVES

Paraschos et al. (2013) introduced a framework for modeling MPs using trajectory distributions, capturing both temporal and inter-dimensional correlations. Unlike DMPs that use a forcing term, ProMPs directly model the intended trajectory. The probability of observing a 1-DoF trajectory $[y_t]_{t=0:T}$ given a specific weight vector distribution $p(\boldsymbol{w}) \sim \mathcal{N}(\boldsymbol{w}|\boldsymbol{\mu_w}, \boldsymbol{\Sigma_w})$ is represented as a linear basis function model:

Linear basis function: $\qquad [y_t]_{t=0:T} = \boldsymbol{\Phi}_{0:T}^{\intercal}\boldsymbol{w} + \epsilon_y,$ (13)

Mapping distribution: $\qquad p([y_t]_{t=0:T}; \boldsymbol{\mu_y}, \boldsymbol{\Sigma_y}) = \mathcal{N}(\boldsymbol{\Phi}_{0:T}^{\intercal}\boldsymbol{\mu_w}, \boldsymbol{\Phi}_{0:T}^{\intercal}\boldsymbol{\Sigma_w}\boldsymbol{\Phi}_{0:T} + \sigma_y^2\boldsymbol{I}).$ (14)

Here, $\epsilon_y$ is zero-mean white noise with variance $\sigma_y^2$. The matrix $\boldsymbol{\Phi}_{0:T}$ houses the basis functions for each time step $t$. Similar to DMPs, these basis functions can be defined in terms of a phase variable instead of time. ProMPs allows for flexible manipulation of MP trajectories through probabilistic operators applied to $p(\boldsymbol{w})$, such as conditioning, combination, and blending (Maeda et al., 2014; Gomez-Gonzalez et al., 2016; Shyam et al., 2019; Rozo & Dave, 2022; Zhou et al., 2019). However, ProMPs lack an intrinsic dynamic system, which means they cannot guarantee a smooth transition from the robot's initial state or between different generated trajectories.

## B.3 PROBABILISTIC DYNAMIC MOVEMENT PRIMITIVES

**Solving the ODE underlying DMPs** Li et al. (2023) noted that the governing equation of DMPs, as specified in Eq. (12), admits an analytical solution. This is because it is a second-order linear non-homogeneous ODE with constant coefficients. The original ODE and its homogeneous counterpart can be expressed in standard form as follows:

Non-homo. ODE: $\quad \ddot{y} + \dfrac{\alpha}{\tau}\dot{y} + \dfrac{\alpha\beta}{\tau^2}y = \dfrac{f(x)}{\tau^2} + \dfrac{\alpha\beta}{\tau^2}g \equiv F(x, g),$ (15)

Homo. ODE: $\quad \ddot{y} + \dfrac{\alpha}{\tau}\dot{y} + \dfrac{\alpha\beta}{\tau^2}y = 0.$ (16)

The solution to this ODE is essentially the position trajectory, and its time derivative yields the velocity trajectory. These are formulated as:

$$y = \begin{bmatrix} y_2\boldsymbol{p_2} - y_1\boldsymbol{p_1} & y_2 q_2 - y_1 q_1 \end{bmatrix} \begin{bmatrix} \boldsymbol{w} \\ g \end{bmatrix} + c_1 y_1 + c_2 y_2 \qquad (17)$$

$$\dot{y} = \begin{bmatrix} \dot{y}_2\boldsymbol{p_2} - \dot{y}_1\boldsymbol{p_1} & \dot{y}_2 q_2 - \dot{y}_1 q_1 \end{bmatrix} \begin{bmatrix} \boldsymbol{w} \\ g \end{bmatrix} + c_1 \dot{y}_1 + c_2 \dot{y}_2. \qquad (18)$$

Here, the learnable parameters $\boldsymbol{w}$ and $g$ which control the shape of the trajectory, are separable from the remaining terms. Time-dependent functions $y_1, y_2, \boldsymbol{p}_1, \boldsymbol{p}_2, q_1, q_2$ in the remaining terms offer the basic support to generate the trajectory. The functions $y_1, y_2$ are the complementary solutions to the homogeneous ODE presented in equation 16, and $\dot{y}_1, \dot{y}_2$ their time derivatives respectively. These time-dependent functions take the form as:

$$y_1(t) = \exp\left(-\frac{\alpha}{2\tau}t\right), \qquad\qquad y_2(t) = t\exp\left(-\frac{\alpha}{2\tau}t\right), \qquad (19)$$

$$\boldsymbol{p}_1(t) = \frac{1}{\tau^2}\int_0^t t'\exp\left(\frac{\alpha}{2\tau}t'\right)x(t')\boldsymbol{\varphi}_x^{\intercal}\mathrm{d}t', \qquad \boldsymbol{p}_2(t) = \frac{1}{\tau^2}\int_0^t \exp\left(\frac{\alpha}{2\tau}t'\right)x(t')\boldsymbol{\varphi}_x^{\intercal}\mathrm{d}t', \qquad (20)$$

$$q_1(t) = \left(\frac{\alpha}{2\tau}t - 1\right)\exp\left(\frac{\alpha}{2\tau}t\right) + 1, \qquad q_2(t) = \frac{\alpha}{2\tau}\left[\exp\left(\frac{\alpha}{2\tau}t\right) - 1\right]. \qquad (21)$$

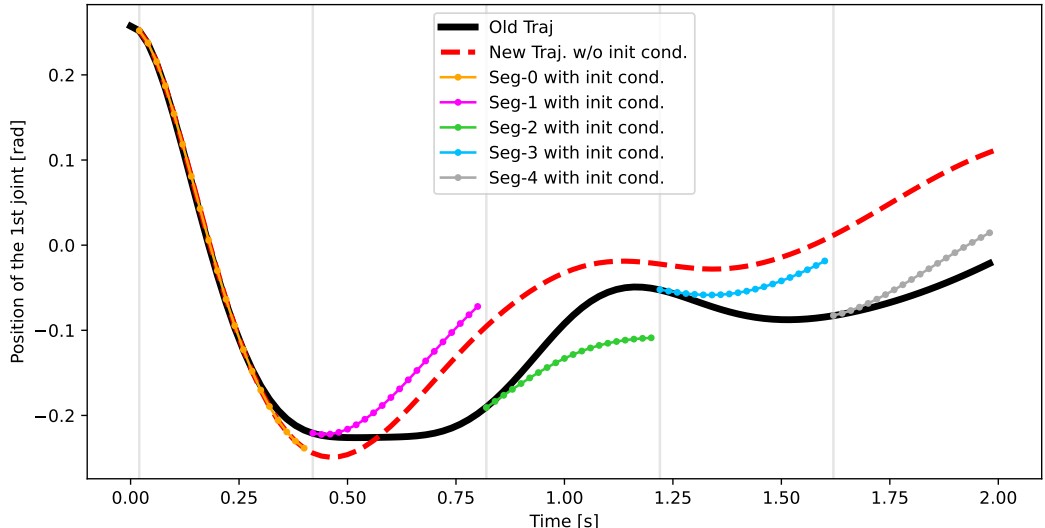

Figure 7: TOP-ERL leverages the initial condition enforcement techniques of a dynamic system to ensure that the new action trajectory starts from the corresponding old state. **These action trajectories are taken from the first DoF of the robot in the box pushing task.**

It's worth noting that the $p_1$ and $p_2$ cannot be analytically derived due to the complex nature of the forcing basis terms $\varphi_x$. As a result, they need to be computed numerically. Despite this, isolating the learnable parameters, namely $w$ and $g$, allows for the reuse of the remaining terms across all generated trajectories. These residual terms can be more specifically identified as the position and velocity basis functions, denoted as $\Phi(t)$ and $\dot{\Phi}(t)$, respectively. When both $w$ and $g$ are included in a concatenated vector, represented as $w_g$, the expressions for position and velocity trajectories can be formulated in a manner akin to that employed by ProMPs:

$$\textbf{Position:} \quad y(t) = \Phi(t)^\intercal w_g + c_1 y_1(t) + c_2 y_2(t), \tag{22}$$

$$\textbf{Velocity:} \quad \dot{y}(t) = \dot{\Phi}(t)^\intercal w_g + c_1 \dot{y}_1(t) + c_2 \dot{y}_2(t). \tag{23}$$

In the main paper, for simplicity and notation convenience, we use $w$ instead of $w_g$ to describe the parameters and goal of ProDMPs.

**Intial Condition Enforcement**  The coefficients $c_1$ and $c_2$ serve as solutions to the initial value problem delineated by the Eq.(22)(23). Li et al. propose utilizing the robot's initial state or the replanning state, characterized by the robot's position and velocity $(y_b, \dot{y}_b)$ to ensure a smooth commencement or transition from a previously generated trajectory. Denote the values of the complementary functions and their derivatives at the condition time $t_b$ as $y_{1_b}, y_{2_b}, \dot{y}_{1_b}$ and $\dot{y}_{2_b}$. Similarly, denote the values of the position and velocity basis functions at this time as $\Phi_b$ and $\dot{\Phi}_b$ respectively. Using these notations, $c_1$ and $c_2$ can be calculated as follows:

$$\begin{bmatrix} c_1 \\ c_2 \end{bmatrix} = \begin{bmatrix} \frac{\dot{y}_{2_b} y_b - y_{2_b} \dot{y}_b}{y_{1_b} \dot{y}_{2_b} - y_{2_b} \dot{y}_{1_b}} + \frac{y_{2_b} \dot{\Phi}_b^\intercal - \dot{y}_{2_b} \Phi_b^\intercal}{y_{1_b} \dot{y}_{2_b} - y_{2_b} \dot{y}_{1_b}} w_g \\ \frac{y_{1_b} \dot{y}_b - \dot{y}_{1_b} y_b}{y_{1_b} \dot{y}_{2_b} - y_{2_b} \dot{y}_{1_b}} + \frac{\dot{y}_{1_b} \Phi_b^\intercal - y_{1_b} \dot{\Phi}_b^\intercal}{y_{1_b} \dot{y}_{2_b} - y_{2_b} \dot{y}_{1_b}} w_g \end{bmatrix}. \tag{24}$$

Despite the complex form used in the initial condition enforcement, the solutions conducted above only rely on solving several linear equations and can be easily implemented in a batch-manner and is therefore computationally efficient, normally $\leq$ 1 ms.

### B.4 ENFORCE TRAJECTORY SEGMENTS' INITIAL CONDITION IN TOP-ERL

In Figure 7, we illustrate how TOP-ERL leverages this mechanism. The figure is based on the motion trajectory of the first degree of freedom of the robot in the box-pushing task. In the critic update, we use five segments as an example.

ProDMP, as a trajectory generator, models the trajectory as a dynamic system. In TOP-ERL, the RL policy predicts ProDMP parameters, which are used to generate a force signal applied to the dynamic system. The system evolves its state based on this force signal and the given initial conditions, such as the robot's position and velocity at a specific time. The resulting evolution trajectory, shown as the black curve in Fig. 7, can be computed in closed form and used to control the robot.

When the policy is updated and predicts a new force signal for the same task, a new action trajectory is generated, depicted as the red dashed curve, which gradually deviates from the old trajectory. However, by utilizing the dynamic system's features, we can set the initial condition of each segment of the new trajectory to the corresponding old state. This ensures that the new action sequence used in the target computation in Eq.(7), can start from the old state, as shown across the five segments in the figure. Therefore, we matched the old state and new actions by eliminating the gap between them, as previously discussed in Section 4.3.1.

From our empirical results, we found that this enforcement is highly beneficial for value function learning. Interestingly, for policy updates, it introduces challenges since these conditions are not used as inputs to the policy. Therefore, we apply this technique only during value function training.

## C EXPERIMENT DETAILS

### C.1 DETAILS OF METHODS IMPLEMENTATION

Table 3: Baseline methods categorized by type (ERL or SRL) and update rules (On- or Off-policy).

| Method | Category | Description |
|---|---|---|
| **BBRL** (Otto et al., 2022) | ERL, On | Black Box Optimization style ERL, policy search in parameter space |
| **TCE** (Li et al., 2024) | ERL, On | Extend BBRL to use per-step info for efficient policy update |
| **PPO** (Schulman et al., 2017) | SRL, On | Standard on-policy method with simplified Trust Region enforcement |
| **gSDE** (Raffin et al., 2022) | SRL, On | Consecutive exploration noise for NN parameters of the policy |
| **GTrXL**(Parisotto et al., 2020) | SRL, On | Transformer-augmented SRL with multiple state as history |
| **SAC** (Haarnoja et al., 2018a) | SRL, Off | Standard off-policy method with entropy bonus for better exploration |
| **PINK** (Eberhard et al., 2022) | SRL, Off | Use temporal correlated pink noise for better exploration |

**PPO** Proximal Policy Optimization (PPO) (Schulman et al., 2017) is a prominent on-policy step-based RL algorithm that refines the policy gradient objective, ensuring policy updates remain close to the behavior policy. PPO branches into two main variants: PPO-Penalty, which incorporates a KL-divergence term into the objective for regularization, and PPO-Clip, which employs a clipped surrogate objective. In this study, we focus our comparisons on PPO-Clip due to its prevalent use in the field. Our implementation of PPO is based on the implementation of Raffin et al. (2021).

**SAC** Soft Actor-Critic (SAC) (Haarnoja et al., 2018a;b) employs a stochastic step-based policy in an off-policy setting and utilizes double Q-networks to mitigate the overestimation of Q-values for stable updates. By integrating entropy regularization into the learning objective, SAC balances between expected returns and policy entropy, preventing the policy from premature convergence. Our implementation of SAC is based on the implementation of Raffin et al. (2021).

**GTrXL** Gated TransformerXL (GTrXL) (Parisotto et al., 2020) is a Transformer architecture that design to stabilize the training of Transformers in online RL, offers an easy-to-train, simple-to-implement but substantially more expressive architectural alternative to standard RNNs used for RL agents in POMDPs. Our implementation of GTrXL is based on the implementation of PPO + GTrXL from Liang et al. (2018). We augmented the implementation with minibatch advantage normalization and state-independent log standard deviation as suggested in Huang et al. (2022).

**gSDE** Generalized State Dependent Exploration (gSDE) (Raffin et al., 2022; Rückstieß et al., 2008; Rückstiess et al., 2010) is an exploration method designed to address issues with traditional

step-based exploration techniques and aims to provide smoother and more efficient exploration in the context of robotic reinforcement learning, reducing jerky motion patterns and potential damage to robot motors while maintaining competitive performance in learning tasks.

To achieve this, gSDE replaces the traditional approach of independently sampling from a Gaussian noise at each time step with a more structured exploration strategy, that samples in a state-dependent manner. The generated samples not only depend on parameter of the Gaussian distribution $\mu$ & $\Sigma$, but also on the activations of the policy network's last hidden layer ($s$). We generate disturbances $\epsilon_t$ using the equation

$$\epsilon_t = \theta_\epsilon s, \text{ where } \theta_\epsilon \sim \mathcal{N}^d(0, \Sigma).$$

The exploration matrix $\theta_\epsilon$ is composed of vectors of length $\text{Dim}(a)$ that were drawn from the Gaussian distribution we want gSDE to follow. The vector $s$ describes how this set of pre-computed exploration vectors are mixed. The exploration matrix is resampled at regular intervals, as guided by the 'sde sampling frequency' (ssf), occurring every n-th step if n is our ssf.

gSDE is versatile, applicable as a substitute for the Gaussian Noise source in numerous on- and off-policy algorithms. We evaluated its performance in an on-policy setting using PPO by utilizing the reference implementation for gSDE from Raffin et al. (2022). In order for training with gSDE to remain stable and reach high performance the usage of a linear schedule over the clip range had to be used for some environments.

**PINK**   We utilize SAC to evaluate the effectiveness of pink noise for efficient exploration. Eberhard et al. (2022) propose to replace the independent action noise $\epsilon_t$ of

$$a_t = \mu_t + \sigma_t \cdot \epsilon_t$$

with correlated noise from particular random processes, whose power spectral density follow a power law. In particular, the use of pink noise, with the exponent $\beta = 1$ in $S(f) = |\mathcal{F}[\epsilon](f)|^2 \propto f^{-\beta}$, should be considered (Eberhard et al., 2022).

We follow the reference implementation and sample chunks of Gaussian pink noise using the inverse Fast Fourier Transform method proposed by Timmer & Koenig (1995). These noise variables are used for SAC's exploration but the the actor and critic updates sample the independent action distribution without pink noise. Each action dimension uses an independent noise process which causes temporal correlation within each dimension but not across dimensions. Furthermore, we fix the chunk size and maximum period to 10000 which avoids frequent jumps of chunk borders and increases relative power of low frequencies.

**BBRL**   Black-Box Reinforcement Learning (BBRL) (Otto et al., 2022; 2023) is a recent developed episodic reinforcement learning method. By utilizing ProMPs (Paraschos et al., 2013) as the trajectory generator, BBRL learns a policy that explores at the trajectory level. The method can effectively handle sparse and non-Markovian rewards by perceiving an entire trajectory as a unified data point, neglecting the temporal structure within sampled trajectories. However, on the other hand, BBRL suffers from relatively low sample efficiency due to its black-box nature. Moreover, the original BBRL employs a degenerate Gaussian policy with diagonal covariance. In this study, we extend BBRL to learn Gaussian policy with full covariance to build a more competitive baseline. For clarity, we refer to the original method as BBRL-Std and the full covariance version as BBRL-Cov. We integrate BBRL with ProDMPs (Li et al., 2023), aiming to isolate the effects attributable to different MP approaches.

**TCE**   Temporally-Correlated Episodic RL (TCE) (Li et al., 2024) is an innovative ERL algorithm that leverages step-level information in episodic policy updates, shedding light on the 'black box' of current ERL methods while preserving smooth and consistent exploration within the parameter space. TCE integrates the strengths of both step-based and episodic RL, offering performance on par with recent ERL approaches, while matching the data efficiency of state-of-the-art (SoTA) step-based RL methods.

## C.2   METAWORLD

MetaWorld (Yu et al., 2020) is an open-source simulated benchmark specifically designed for meta-reinforcement learning and multi-task learning in robotic manipulation. It features 50 distinct ma-

nipulation tasks, each presenting unique challenges that require robots to learn a wide range of skills, such as grasping, pushing, and object placement. Unlike benchmarks that focus on narrow task distributions, MetaWorld provides a broader range of tasks, making it an ideal platform for developing algorithms that can generalize across different behaviors.

To ensure a fair comparison, we followed the evaluation protocol described in Otto et al. (2022) and Li et al. (2024), where an episode is considered successful only if the success criterion is met at the end of the episode. This is equivalent to requiring the robot to complete the task and maintain its success state until the episode ends, which is a more rigorous measure than the original setting, where success at any time step is sufficient.

We reported each individual Metaworld task in Fig. 8 and Fig. 9. These tasks cover a wide range of types and complexities.

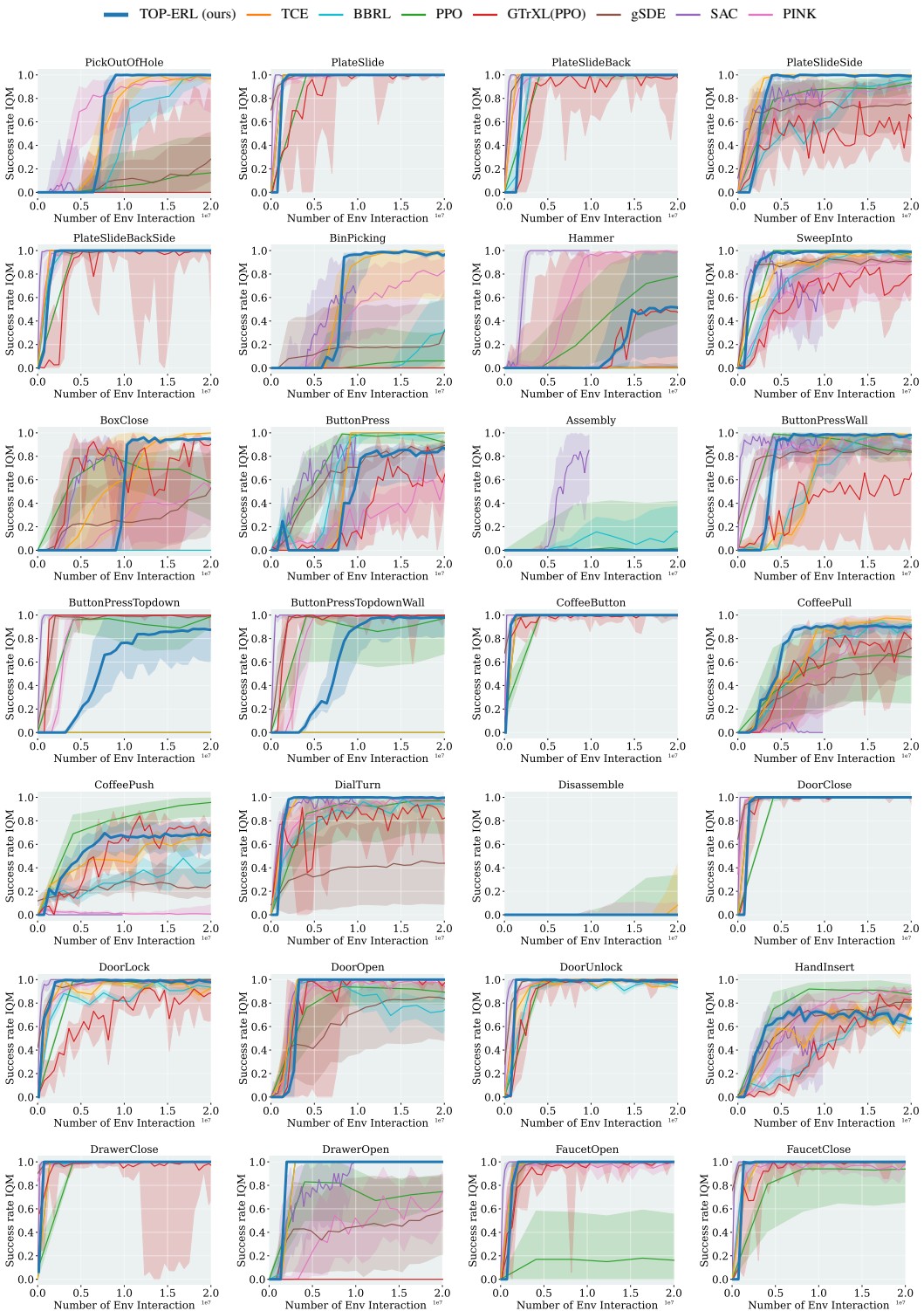

Figure 8: Success Rate IQM of each individual Metaworld tasks.

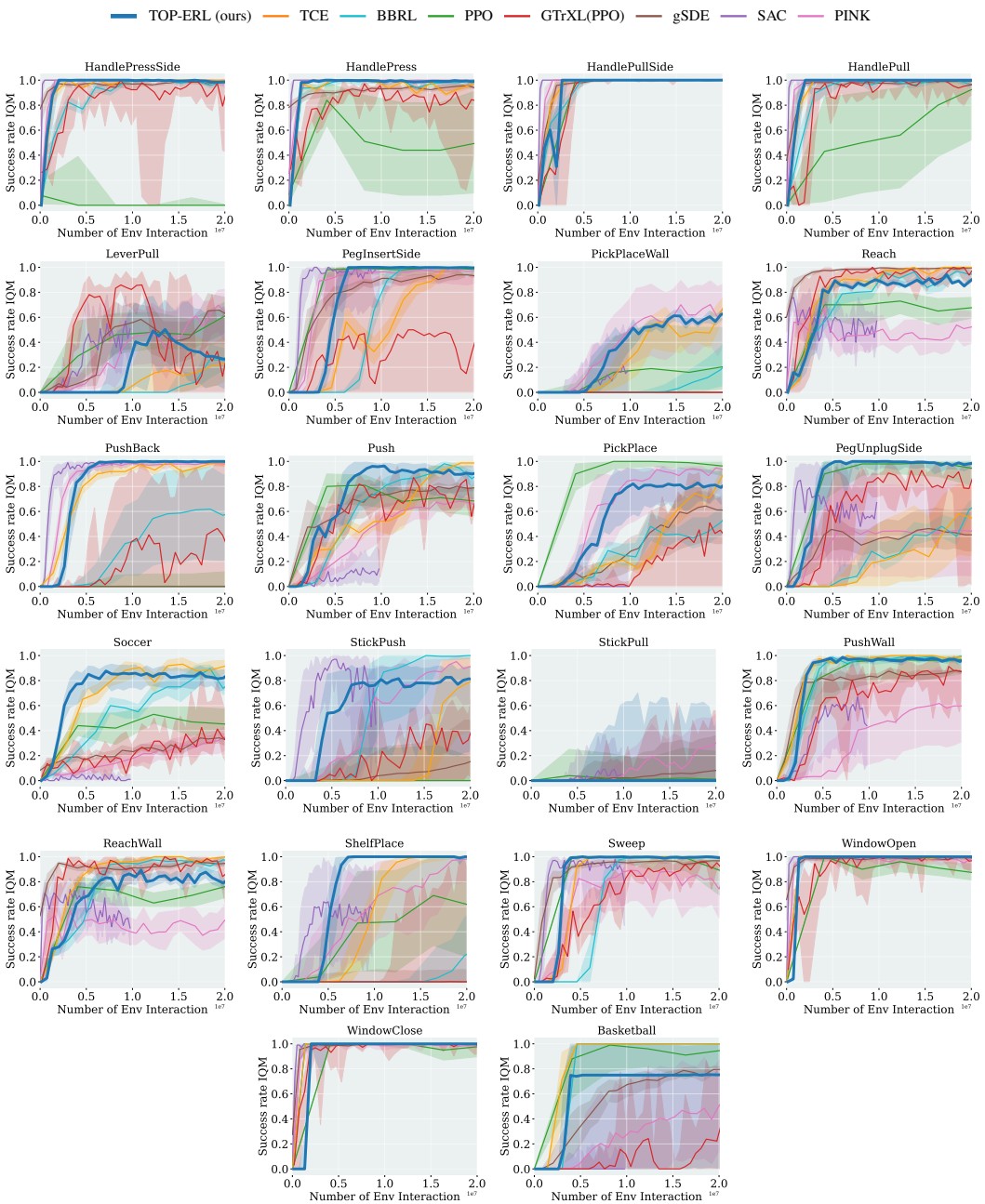

Figure 9: Success Rate IQM of each individual Metaworld tasks.

## C.3 HOPPER JUMP

As an addition to the main paper, we provide more details on the Hopper Jump task. We look at both the main goal of maximizing jump height and the secondary goal of landing on a desired position. Our method shows quick learning and does well in achieving high jump height, consistent with what we reported earlier. While it's not as strong in landing accuracy, it still ranks high in overall performance. Both versions of BBRL have similar results. However, they train more slowly compared to TCE, highlighting the speed advantage of our method due to the use of intermediate states for policy updates. Looking at other methods, step-based ones like PPO and TRPL focus too much on landing distance and miss out on jump height, leading to less effective policies. On the other hand, gSDE performs well but is sensitive to the initial setup, as shown by the

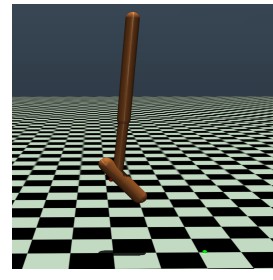

Figure 10: Hopper Jump

wide confidence ranges in the results. Lastly, SAC and PINK shows inconsistent results in jump height, indicating the limitations of using pink noise for exploration, especially when compared to gSDE.

## C.4 BOX PUSHING

The goal of the box-pushing task is to move a box to a specified goal location and orientation using the 7-DoFs Franka Emika Panda (Otto et al., 2022). To make the environment more challenging, we extend the environment from a fixed initial box position and orientation to a randomized initial position and orientation. The range of both initial and target box pose varies from $x \in [0.3, 0.6], y \in [-0.45, 0.45], \theta_z \in [0, 2\pi]$. Success is defined as a positional distance error of less than 5 cm and a z-axis orientation error of less than 0.5 rad. We refer to the original paper for the observation and action spaces definition and the reward function.

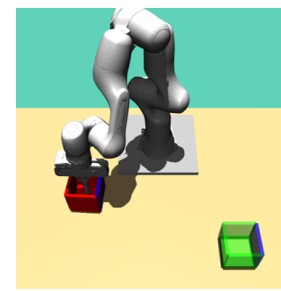

Figure 11: Box Pushing

# D ACTION CORRELATION WITH SEGMENTATION

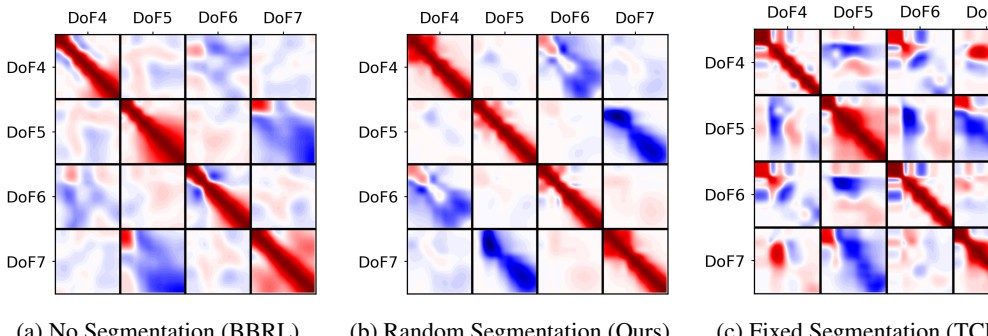

| (a) No Segmentation (BBRL) | (b) Random Segmentation (Ours) | (c) Fixed Segmentation (TCE) |

Figure 12: This figure presents predicted actions' correlation across 4 DoF and 100 time steps, visualized in a $400 \times 400$ correlation matrix. Each $100 \times 100$ square tile demonstrates the movement correlation between two DoF during these steps. Correlation values range from -1 (negative correlation, depicted in blue) to 1 (positive correlation, depicted in red), with white areas indicating no correlation. BBRL treats the entire trajectory as a whole and does not have any segmentation; thus, the correlation broadcasts smoothly across time steps, as shown in (a). On the contrary, TCE uses segmentation with fixed length, constraining the correlation learning within fixed segments, resulting in sudden correlation changes at each segment's boundary, as presented in (c). TOP-ERL utilizes randomly sampled segment length and positions itself between the two paradigms, being able to learn the smooth correlation while retaining the benefits of higher sample efficiency by using segmentation.

## E  HYPER PARAMETERS

We executed a large-scale grid search to fine-tune key hyperparameters for each baseline method. For other hyperparameters, we relied on the values specified in their respective original papers. Below is a list summarizing the parameters we swept through during this process.

**BBRL:**  Policy net size, critic net size, policy learning rate, critic learning rate, samples per iteration, trust region dissimilarity bounds, number of parameters per movement DoF.

**TCE:**  Same types of hyper-parameters listed in BBRL, plus the number of segments per trajectory. A learning rate decaying scheduler is applied to stabilize the training in the end.

**PPO:**  Policy network size, critic network size, policy learning rate, critic learning rate, batch size, samples per iteration.

**gSDE:**  Same types of hyper-parameters listed in PPO, together with the state dependent exploration sampling frequency (Raffin et al., 2022).

**SAC:**  Policy network size, critic network size, policy learning rate, critic learning rate, alpha learning rate, batch size, Update-To-Data (UTD) ratio.

**PINK:**  Same types of hyper-parameters listed in SAC.

**GTrXL:**  Number of multi-head attention layers, number of heads, dims per head, importance-sampling ratio clip, value function clip, grad clip, and same hyperparameters listed in PPO

**TOP-ERL:**  Number of multi-head attention layers, number of heads, dims per head, learning rates. The other movement primitives hyper-parameters are taken from TCE.

The detailed hyper parameters used are listed in the following tables. Unless stated otherwise, the notation lin_x refers to a linear schedule. It interpolates linearly from x to 0 during training. The ERL methods (TCE, BBRL) take an entire trajectory as a sample where the SRL methods take one time step as a sample. In this way, one sample in ERL is equivlent to $T$ sample of SRL, where $T$ is the length of one task episode.

Table 4: Hyperparameters for the Meta-World experiments. Episode Length $T = 500$

| | PPO | gSDE | GTrXL | SAC | PINK | TCE | BBRL | TOP-ERL |
|---|---|---|---|---|---|---|---|---|
| number samples | 16000 | 16000 | 19000 | 1000 | 4 | 16 | 16 | 2 |
| GAE $\lambda$ | 0.95 | 0.95 | 0.95 | n.a. | n.a. | 0.95 | n.a. | n.a. |
| discount factor | 0.99 | 0.99 | 0.99 | 0.99 | 0.99 | 1 | 1 | 1.0 |
| | | | | | | | | |
| $\epsilon_\mu$ | n.a. | n.a. | n.a. | n.a. | n.a. | 0.005 | 0.005 | 0.005 |
| $\epsilon_\Sigma$ | n.a. | n.a. | n.a. | n.a. | n.a. | 0.0005 | 0.0005 | 0.0005 |
| trust region loss coef. | n.a. | n.a. | n.a. | n.a. | n.a. | 1 | 10 | 1.0 |
| | | | | | | | | |
| optimizer | adam | adam | adam | adam | adam | adam | adam | adam |
| epochs | 10 | 10 | 5 | 1000 | 1 | 50 | 100 | 15 |
| learning rate | 3e-4 | 1e-3 | 2e-4 | 3e-4 | 3e-4 | 3e-4 | 3e-4 | 1e-3 |
| use critic | True | True | True | True | True | True | True | True |
| epochs critic | 10 | 10 | 5 | 1000 | 1 | 50 | 100 | 50 |
| learning rate critic | 3e-4 | 1e-3 | 2e-4 | 3e-4 | 3e-4 | 3e-4 | 3e-4 | 5e-5 |
| number minibatches | 32 | n.a. | n.a | n.a. | n.a. | n.a. | n.a. | n.a. |
| batch size | n.a. | 500 | 1024 | 256 | 512 | n.a. | n.a. | 256 |
| buffer size | n.a. | n.a. | n.a. | 1e6 | 2e6 | n.a. | n.a. | 3000 |
| learning starts | 0 | 0 | n.a. | 10000 | 1e5 | 0 | 0 | 2 |
| polyak_weight | n.a. | n.a. | n.a. | 5e-3 | 5e-3 | n.a. | n.a. | 5e-3 |
| SDE sampling frequency | n.a. | 4 | n.a. | n.a. | n.a. | n.a. | n.a. | n.a. |
| entropy coefficient | 0 | 0 | 0 | auto | auto | 0 | 0 | n.a. |
| | | | | | | | | |
| normalized observations | True | True | False | False | False | True | False | False |
| normalized rewards | True | True | 0.05 | False | False | False | False | False |
| observation clip | 10.0 | n.a. | n.a. | n.a. | n.a. | n.a. | n.a. | n.a. |
| reward clip | 10.0 | 10.0 | 10.0 | n.a. | n.a. | n.a. | n.a. | n.a. |
| critic clip | 0.2 | lin_0.3 | 10.0 | n.a. | n.a. | n.a. | n.a. | n.a. |
| importance ratio clip | 0.2 | lin_0.3 | 0.1 | n.a. | n.a. | n.a. | n.a. | n.a. |
| | | | | | | | | |
| hidden layers | [128, 128] | [128, 128] | n.a. | [256, 256] | [256, 256] | [128, 128] | [32, 32] | [ 128, 128] |
| hidden layers critic | [128, 128] | [128, 128] | n.a. | [256, 256] | [256, 256] | [128, 128] | [32, 32] | n.a. |
| hidden activation | tanh | tanh | relu | relu | relu | relu | relu | leaky_relu |
| orthogonal initialization | Yes | No | xavier | fanin | fanin | Yes | Yes | Yes |
| initial std | 1.0 | 0.5 | 1.0 | 1.0 | 1.0 | 1.0 | 1.0 | 1.0 |
| number of heads | - | - | 4 | - | - | - | - | 8 |
| dims per head | - | - | 16 | - | - | - | - | 16 |
| number of attention layers | - | - | 4 | - | - | - | - | 2 |
| max sequence length | - | - | 5 | - | - | - | - | 1024 |

---

[1]Linear Schedule from 0.3 to 0.01 during the first 25% of the training. Then continued with 0.01.

Table 5: Hyperparameters for the Box Pushing Dense, Episode Length $T = 100$

|  | PPO | gSDE | GTrXL | SAC | PINK | TCE | BBRL | TOP-ERL |
|---|---|---|---|---|---|---|---|---|
| number samples | 48000 | 80000 | 8000 | 8 | 8 | 152 | 152 | 4 |
| GAE $\lambda$ | 0.95 | 0.95 | 0.95 | n.a. | n.a. | 0.95 | n.a. | n.a. |
| discount factor | 1.0 | 1.0 | 0.99 | 0.99 | 0.99 | 1.0 | 1.0 | 1.0 |
| $\epsilon_\mu$ | n.a. | n.a. | n.a. | n.a. | n.a. | 0.05 | 0.1 | 0.005 |
| $\epsilon_\Sigma$ | n.a. | n.a. | n.a. | n.a. | n.a. | 0.0005 | 0.00025 | 0.0005 |
| trust region loss coef. | n.a. | n.a. | n.a. | n.a. | n.a. | 1 | 10 | 1.0 |
| optimizer | adam | adam | adam | adam | adam | adam | adam | adam |
| epochs | 10 | 10 | 5 | 1 | 1 | 50 | 20 | 15 |
| learning rate | 5e-5 | 1e-4 | 2e-4 | 3e-4 | 3e-4 | 3e-4 | 3e-4 | 3e-4 |
| use critic | True | True | True | True | True | True | True | True |
| epochs critic | 10 | 10 | 5 | 1 | 1 | 50 | 10 | 30 |
| learning rate critic | 1e-4 | 1e-4 | 2e-4 | 3e-4 | 3e-4 | 1e-3 | 3e-4 | 5e-5 |
| number minibatches | 40 | n.a. | n.a. | n.a. | n.a. | n.a. | n.a. | n.a. |
| batch size | n.a. | 2000 | 1000 | 512 | 512 | n.a. | n.a. | 512 |
| buffer size | n.a. | n.a. | n.a. | 2e6 | 2e6 | n.a. | n.a. | 7000 |
| learning starts | 0 | 0 | 0 | 1e5 | 1e5 | 0 | 0 | 8000 |
| polyak_weight | n.a. | n.a. | n.a. | 5e-3 | 5e-3 | n.a. | n.a. | 5e-3 |
| SDE sampling frequency | n.a. | 4 | n.a. | n.a. | n.a. | n.a. | n.a. | n.a. |
| entropy coefficient | 0 | 0.01 | 0 | auto | auto | 0 | 0 | 0. |
| normalized observations | True | True | False | False | False | True | False | False |
| normalized rewards | True | True | 0.1 | False | False | False | False | False |
| observation clip | 10.0 | n.a. | n.a. | n.a. | n.a. | n.a. | n.a. | n.a. |
| reward clip | 10.0 | 10.0 | 10. | n.a. | n.a. | n.a. | n.a. | n.a. |
| critic clip | 0.2 | 0.2 | 10. | n.a. | n.a. | n.a. | n.a. | n.a. |
| importance ratio clip | 0.2 | 0.2 | 0.1 | n.a. | n.a. | n.a. | n.a. | n.a. |
| hidden layers | [512, 512] | [256, 256] | n.a. | [256, 256] | [256, 256] | [128, 128] | [128, 128] | [256, 256] |
| hidden layers critic | [512, 512] | [256, 256] | n.a. | [256, 256] | [256, 256] | [256, 256] | [256, 256] | n.a. |
| hidden activation | tanh | tanh | relu | tanh | tanh | leaky_relu | leaky_relu | leaky_relu |
| orthogonal initialization | Yes | No | xavier | fanin | fanin | Yes | Yes | Yes |
| initial std | 1.0 | 0.05 | 1.0 | 1.0 | 1.0 | 1.0 | 1.0 | 1.0 |
| number of heads | - | - | 4 | - | - | - | - | 8 |
| dims per head | - | - | 16 | - | - | - | - | 16 |
| number of attention layers | - | - | 4 | - | - | - | - | 2 |
| max sequence length | - | - | 5 | - | - | - | - | 1024 |
| Movement Primitive (MP) type | n.a. | n.a. | value | n.a. | n.a. | ProDMPs | ProDMPs | ProDMPs |
| number basis functions | n.a. | n.a. | value | n.a. | n.a. | 8 | 8 | 8 |
| weight scale | n.a. | n.a. | value | n.a. | n.a. | 0.3 | 0.3 | 0.3 |
| goal scale | n.a. | n.a. | value | n.a. | n.a. | 0.3 | 0.3 | 0.3 |

Table 6: Hyperparameters for the Box Pushing Sparse, Episode Length $T = 100$

|  | PPO | gSDE | GTrXL | SAC | PINK | TCE | BBRL | TOP-ERL |
|---|---|---|---|---|---|---|---|---|
| number samples | 48000 | 80000 | 8000 | 8 | 8 | 76 | 76 | 4 |
| GAE $\lambda$ | 0.95 | 0.95 | 0.95 | n.a. | n.a. | 0.95 | n.a. | n.a. |
| discount factor | 1.0 | 1.0 | 1.0 | 0.99 | 0.99 | 1.0 | 1.0 | 1.0 |
| | | | | | | | | |
| $\epsilon_\mu$ | n.a. | n.a. | n.a. | n.a. | n.a. | 0.05 | 0.1 | 0.005 |
| $\epsilon_\Sigma$ | n.a. | n.a. | n.a. | n.a. | n.a. | 0.0005 | 0.00025 | 0.0005 |
| trust region loss coef. | n.a. | n.a. | n.a. | n.a. | n.a. | 1 | 10 | 1.0 |
| | | | | | | | | |
| optimizer | adam | adam | adam | adam | adam | adam | adam | adam |
| epochs | 10 | 10 | 5 | 1 | 1 | 50 | 20 | 15 |
| learning rate | 5e-4 | 1e-4 | 2e-4 | 3e-4 | 3e-4 | 3e-4 | 3e-4 | 3e-4 |
| use critic | True | True | True | True | True | True | True | True |
| epochs critic | 10 | 10 | 5 | 1 | 1 | 50 | 10 | 30 |
| learning rate critic | 1e-4 | 1e-4 | 2e-4 | 3e-4 | 3e-4 | 3e-4 | 3e-4 | 5e-5 |
| number minibatches | 40 | n.a. | n.a. | n.a. | n.a. | n.a. | n.a. | n.a. |
| batch size | n.a. | 2000 | 1000 | 512 | 512 | n.a. | n.a. | 512 |
| buffer size | n.a. | n.a. | n.a. | 2e6 | 2e6 | n.a. | n.a. | 7000 |
| learning starts | 0 | 0 | 0 | 1e5 | 1e5 | 0 | 0 | 400 |
| polyak_weight | n.a. | n.a. | 0 | 5e-3 | 5e-3 | n.a. | n.a. | 5e-3 |
| SDE sampling frequency | n.a. | 4 | 0 | n.a. | n.a. | n.a. | n.a. | n.a. |
| entropy coefficient | 0 | 0.01 | 0 | auto | auto | 0 | 0 | 0 |
| | | | | | | | | |
| normalized observations | True | True | False | False | False | True | False | False |
| normalized rewards | True | True | 0.1 | False | False | False | False | False |
| observation clip | 10.0 | n.a. | False | n.a. | n.a. | n.a. | n.a. | n.a. |
| reward clip | 10.0 | 10.0 | 10.0 | n.a. | n.a. | n.a. | n.a. | n.a. |
| critic clip | 0.2 | 0.2 | 10.0 | n.a. | n.a. | n.a. | n.a. | n.a. |
| importance ratio clip | 0.2 | 0.2 | 0.1 | n.a. | n.a. | n.a. | n.a. | n.a. |
| | | | | | | | | |
| hidden layers | [512, 512] | [256, 256] | n.a. | [256, 256] | [256, 256] | [128, 128] | [128, 128] | [256, 256] |
| hidden layers critic | [512, 512] | [256, 256] | n.a. | [256, 256] | [256, 256] | [256, 256] | [256, 256] | n.a. |
| hidden activation | tanh | tanh | relu | tanh | tanh | leaky_relu | leaky_relu | leaky_relu |
| orthogonal initialization | Yes | No | xavier | fanin | fanin | Yes | Yes | Yes |
| initial std | 1.0 | 0.05 | 1.0 | 1.0 | 1.0 | 1.0 | 1.0 | 1.0 |
| number of heads | - | - | 4 | - | - | - | - | 8 |
| dims per head | - | - | 16 | - | - | - | - | 16 |
| number of attention layers | - | - | 4 | - | - | - | - | 2 |
| max sequence length | - | - | 5 | - | - | - | - | 1024 |
| | | | | | | | | |
| MP type | n.a. | n.a. | value | n.a. | n.a. | ProDMPs | ProDMPs | ProDMPs |
| number basis functions | n.a. | n.a. | value | n.a. | n.a. | 8 | 8 | 8 |
| weight scale | n.a. | n.a. | value | n.a. | n.a. | 0.3 | 0.3 | 0.3 |
| goal scale | n.a. | n.a. | value | n.a. | n.a. | 0.3 | 0.3 | 0.3 |

Table 7: Hyperparameters for the Hopper Jump, Episode Length $T = 250$

| | PPO | gSDE | GTrXL | SAC | PINK | TCE | BBRL | TOP-ERL |
|---|---|---|---|---|---|---|---|---|
| number samples | 8000 | 8192 | 10000 | 1000 | 1 | 64 | 64 | 1 |
| GAE $\lambda$ | 0.95 | 0.99 | 0.95 | n.a. | n.a. | 0.95 | n.a. | n.a. |
| discount factor | 1.0 | 0.999 | 1.0 | 0.99 | 0.99 | 1.0 | 1.0 | 1.0 |
| | | | | | | | | |
| $\epsilon_\mu$ | n.a. | n.a. | n.a. | n.a. | n.a. | 0.1 | n.a. | 0.1 |
| $\epsilon_\Sigma$ | n.a. | n.a. | n.a. | n.a. | n.a. | 0.02 | n.a. | 0.02 |
| trust region loss coef. | n.a. | n.a. | n.a. | n.a. | n.a. | 1 | n.a. | 1.0 |
| | | | | | | | | |
| optimizer | adam | adam | adam | adam | adam | adam | adam | adam |
| epochs | 10 | 10 | 10 | 1000 | 1 | 50 | 100 | 10 |
| learning rate | 3e-4 | 9.5e-5 | 5e-4 | 1e-4 | 2e-4 | 1e-4 | 1e-4 | 1e-4 |
| use critic | True | True | True | True | True | True | True | True |
| epochs critic | 10 | 10 | 10 | 1000 | 1 | 50 | 100 | 20 |
| learning rate critic | 3e-4 | 9.5e-5 | 5e-4 | 1e-4 | 2e-4 | 1e-4 | 1e-4 | 5e-5 |
| number minibatches | 40 | n.a. | n.a. | n.a. | n.a. | n.a. | n.a. | n.a. |
| batch size | n.a. | 128 | 1024 | 256 | 256 | n.a. | n.a. | 256 |
| buffer size | n.a. | n.a. | n.a. | 1e6 | 1e6 | n.a. | n.a. | 1000 |
| learning starts | 0 | 0 | 0 | 10000 | 1e5 | 0 | 0 | 250 |
| polyak_weight | n.a. | n.a. | n.a. | 5e-3 | 5e-3 | n.a. | n.a. | 5e-3 |
| SDE sampling frequency | n.a. | 8 | n.a. | n.a. | n.a. | n.a. | n.a. | n.a. |
| entropy coefficient | 0 | 0.0025 | 0. | auto | auto | 0 | 0 | 0 |
| | | | | | | | | |
| normalized observations | True | False | False | False | False | True | False | False |
| normalized rewards | True | False | False | False | False | False | False | False |
| observation clip | 10.0 | n.a. | False | n.a. | n.a. | n.a. | n.a. | n.a. |
| reward clip | 10.0 | 10.0 | 10. | n.a. | n.a. | n.a. | n.a. | n.a. |
| critic clip | 0.2 | lin_0.4 | 1. | n.a. | n.a. | n.a. | n.a. | n.a. |
| importance ratio clip | 0.2 | lin_0.4 | 0.2 | n.a. | n.a. | n.a. | n.a. | n.a. |
| | | | | | | | | |
| hidden layers | [32, 32] | [256, 256] | n.a. | [256, 256] | [32, 32] | [128, 128] | [32, 32] | [128, 128] |
| hidden layers critic | [32, 32] | [256, 256] | n.a | [256, 256] | [32, 32] | [128, 128] | [32, 32] | n.a. |
| hidden activation | tanh | tanh | relu | relu | relu | leaky_relu | tanh | leaky_relu |
| orthogonal initialization | Yes | No | xavier | fanin | fanin | Yes | Yes | Yes |
| initial std | 1.0 | 0.1 | 1.0 | 1.0 | 1.0 | 1.0 | 1.0 | 1.0 |
| number of heads | - | - | 4 | - | - | - | - | 8 |
| dims per head | - | - | 16 | - | - | - | - | 16 |
| number of attention layers | - | - | 4 | - | - | - | - | 2 |
| max sequence length | - | - | 5 | - | - | - | - | 1024 |
| | | | | | | | | |
| MP type | n.a. | n.a. | value | n.a. | n.a. | ProDMPs | ProDMPs | ProDMPs |
| number basis functions | n.a. | n.a. | value | n.a. | n.a. | 3 | 3 | 3 |
| weight scale | n.a. | n.a. | value | n.a. | n.a. | 1 | 1 | 1 |
| goal scale | n.a. | n.a. | value | n.a. | n.a. | 1 | 1 | 1 |

