# OpenReview forum: "TOP-ERL: Transformer-based Off-Policy Episodic Reinforcement Learning"
_ICLR.cc/2025/Conference — ICLR 2025 Spotlight_

### Official Review · Reviewer_L7MP · 2024-10-29

**Soundness:** 3
**Presentation:** 4
**Contribution:** 3
**Rating:** 8
**Confidence:** 4

**Summary:**

This paper presented a transformer-based episodic reinforcement learning method for enabling online updates in an off-policy RL framework. Combined by a trajectory segmentation technique, the transformer architecture is employed to construct a critic that evaluates the values of segments of state-action trajectories. As a result, the proposed method achieves an off-policy update, improving update efficiency compared to previous episodic RL methods using on-policy training. Empirical evaluations on control tasks show that the proposed method improves performance.

**Strengths:**

- The paper is well motivated.
- The writing is quite good.
- The main experiments are conducted on the large-scale RL benchmark.
- Analysis of many design choices are presented.

**Weaknesses:**

- Some used techniques, such as trust region constraints, layer normalization, greatly affect the performance of the proposed algorithm (see Fig. 5). The ablations of TOP-ERL, which do not use these techniques, actually achieved worse performance than previous methods.  Someone may conclude that the performance improvement of TOP-ERL over baselines is achieved by using all these techniques, rather than the usage of the transformer architecture or others.

- Layer normalization and trust region constraints is important for TOP-ERL achieving good performance. But the paper lacks a detailed discussion about the importance of these two components.

**Questions:**

-  Designing additional experiments to verify the individual contribution of the transformer-based critic network, such as TOP-ERL without the transformer, is suggested. However, I have no idea about the specific details about the exp.
- Explaining why layer normalization and trust region constraints are critical in Section 5.3 are suggested.

---

> ### Author Response · Authors · 2024-11-18
> **Reply to reviewer L7MP**
>
> Dear Reviewer L7MP,
>
> Thank you for your insightful and postive review of our paper. We have addressed your concerns as follows and included some detailed discussion in the supplementary material.
>
> **Update Log [Nov. 18]**
>
> - Address the concerns of all reviewers.
> - Upload additional experiment and ablation study to Supplementary Material.
>
>     Link: https://openreview.net/attachment?id=N4NhVN30ph&name=supplementary_material
>
> - Revise the manuscript, with modifications marked in blue.
>
>     Link: https://openreview.net/pdf?id=N4NhVN30ph
>
> =======================================================
>
> ## More details regarding TRPL? and LayerNorm?
>
> > Some used techniques, such as trust region constraints, layer normalization, greatly affect the performance of the proposed algorithm (see Fig. 5). The ablations of TOP-ERL, which do not use these techniques, actually achieved worse performance than previous methods. Someone may conclude that the performance improvement of TOP-ERL over baselines is achieved by using all these techniques, rather than the usage of the transformer architecture or others.
> >
>
> Thank you for pointing out this issue. We briefly introduce these two techniques in this reply and have added a comprehensive discussion of TRPL in Section R3 of the supplementary material, which we plan to include as an additional appendix section of our paper in the future.
>
> **TRPL**
>
> The TRPL [1] method enforces divergence constraints, such as KL-divergence, between the old and new Gaussian policies after each policy update iteration. Unlike the likelihood clipping method commonly used in PPO [2], TRPL constrains both the scale and rotation of the conditional Gaussian distribution $\pi(a|s)$ at each state $s$ in the minibatch, ensuring stable learning. If an updated policy goes beyond the trust region, TRPL projects its Gaussian parameters back to the trust region boundary and adds a trust region regression loss to the RL learning objective. This capability is essential for policies that use high-dimensional Gaussian distributions with a full covariance matrix. Previous episodic RL methods, such as [3-5], utilize TRPL for stable training, with a comprehensive ablation comparing TRPL and likelihood clipping available in [4]. We provide more details in the supplementary in Section R3.
>
> **LayerNorm**
>
> Layernorm[6] and dropout are two standard techniques utilized in the original transformer paper [7], where they were shown to be effective for sequence modeling, such as language models. In TOP-ERL, we examined these techniques and found that retaining layernorm in the transformer critic is beneficial, as it provides the only normalization in our model. However, applying dropout harms performance. We attribute this to the nature of RL data, which is generated through agent-environment interactions and inherently contains randomness. Unlike in supervised learning, RL models are less prone to overfitting; thus, adding additional randomness through dropout makes value prediction more challenging.
>
> ## Design addtional ablation studies to verify the contribution of the Transformer critic?
>
> > Designing additional experiments to verify the individual contribution of the transformer-based critic network, such as TOP-ERL without the transformer, is suggested. However, I have no idea about the specific details about the exp.
> >
>
> We thank the reviewer for this valuable suggestion to help analyze the effectiveness of our model. In response, we conducted two additional ablation studies (introduced below) to address this concern, with their performance results detailed in Section R6 of the supplementary material. However, both ablated methods demonstrated significantly poorer performance compared to the default setting of TOP-ERL.
>
> **SAC + MP**
>
> The approach involves directly using SAC [8] to predict the MP’s parameters while retaining the trajectory generation and environment rollout procedure described in Section 4.1. We sum up the rewards collected during trajectory execution and use this as the return for the entire trajectory. Here, the trajectory is treated as a single action, and the learning objective does not incorporate any temporal information within the trajectory.
>
> **Degenerate Critic from Taking Sequence of Actions to a Single Action**
>
> The approach retains the main architecture of TOP-ERL but reduces the segment length to one in critic updates. In this case, the transformer critic degenerates to a standard critic network, as the input consists of only one state and one action rather than a sequence of actions.
>
> (To be continued...)

---

> ### Author Response · Authors · 2024-11-18
> **Reply to reviewer L7MP (part 2)**
>
> (Continue from part 1...)
>
> **Reference**
>
> [1] Otto, Fabian, et al. "Differentiable trust region layers for deep reinforcement learning." ICLR 2021.
>
> [2] Schulman, John, et al. "Proximal policy optimization algorithms." *arXiv preprint arXiv:1707.06347* (2017).
>
> [3] Onur Celik, et al. "Acquiring Diverse Skills using Curriculum Reinforcement Learning with Mixture of Experts.", ICML 2024. **
>
> [4] Otto, Fabian, et al. "Deep black-box reinforcement learning with movement primitives." *Conference on Robot Learning*. PMLR, 2022.
>
> [5] Li Ge, et al. “Open the Black Box: Step-based Policy Updates for Temporally-Correlated Episodic Reinforcement Learning.” ICLR 2024.
>
> [6] Ba, Jimmy Lei. "Layer normalization." NIPS 2016 Deep Learning Symposium
>
> [7] Vaswani, A. "Attention is all you need." NIPS 2017.
>
> [8] Haarnoja, Tuomas, et al. "Soft actor-critic: Off-policy maximum entropy deep reinforcement learning with a stochastic actor." ICML 2018.
>
> ================================================
>
> If you have any further questions, please feel free to ask us.
>
> Thank you.
>
> TOP-ERL Authors

---

> ### Author Response · Authors · 2024-11-25
> **We look forward to your response.**
>
> Dear Reviewer L7MP,
>
> As the rebuttal period nears its end, we would appreciate knowing if our responses have sufficiently addressed your concerns. Your feedback will assist us in making a more informed decision about our submission. Thank you for your time and consideration.
>
> Best,
>
> TOP-ERL Authors

---

> > ### Comment · Reviewer_L7MP · 2024-11-25
> > **Comment by Reviewer L7MP**
> >
> > Thank you for solving my all concerns. I have increased my score in response.

---

> > > ### Author Response · Authors · 2024-11-25
> > > **Thank you**
> > >
> > > Dear Reviewer,
> > >
> > > We greatly appreciate your approval of our work. Thank you!
> > >
> > > Best regards,
> > >
> > > TOP-ERL Authors

---

### Official Review · Reviewer_iE4K · 2024-11-01

**Soundness:** 3
**Presentation:** 4
**Contribution:** 3
**Rating:** 8
**Confidence:** 4

**Summary:**

The paper introduces Transformer-based Off-Policy Episodic Reinforcement Learning (TOP-ERL), an interesting combination of episodic RL with off-policy updates and the transformer architecture.
The main contribution is the way to use the transformer architecture in this setting to learn an off-policy critic.
The authors leverage N-step returns, Trust Region Projection Layer, Probabilistic Movement Primitives and Target Networks.
Empirical results on robot learning environments (Meta-World, etc.) show that TOP-ERL outperforms state-of-the-art RL methods, both in terms of policy quality and sample efficiency. The paper also includes thorough ablation studies to analyze the impact of key design choices.

**Strengths:**

I find the paper very well written, also well structured, motivated and explained.

The idea of using a transformer such that s_0 maps to V(s), a_0 maps to Q(s, a_0), a_1 maps to Q(s, a_1), ... is very natural and creative.
To the best of my knowledge, it is the first approach that proposes to do it this way.

I also find it interesting that V and Q can be modeled by the same unique transformer architecture without having dedicated weight just for V or just for Q.

The contributions of this paper are good for the field of RL with transformers, although I believe ERL is slightly less used than standard RL.

**Weaknesses:**

The authors claim that off-policy algorithms are often more sample efficient than on-policy counterparts. However, PPO is still often the way to go when you face a new RL problems because of better stability and the need for less hyperparameter tuning. I am not fully convinced this approach won't have the same issue.

I believe the explanations of why importance sampling is not necessary could be improved.

**Questions:**

1. It is not completely clear to me if the transformers can attend only the current observation s^k_0 or also the previous ones during training?

2. I am wondering how important are the probabilistic MP in the performance, what if the policy was also a transformer?

3. I am also wondering if the positional encoding of two different states s^k_0 and s'^k_0 would always be the same (even if one is more in the future that the other one)?

4. Is a target network necessary at all?

---

> ### Author Response · Authors · 2024-11-18
> **Reply to reviewer iE4K**
>
> **Dear Reviewer iE4K**
>
> We appreciate your positive review of our paper and are glad that our paper’s novelty and contributions are well-understood. We address your concerns as follows to the best of our knowledge.
>
> **Update Log [Nov. 18]**
>
> - Address the concerns of all reviewers.
> - Upload additional experiment and ablation study to Supplementary Material.
>
>     Link: https://openreview.net/attachment?id=N4NhVN30ph&name=supplementary_material
>
> - Revise the manuscript, with modifications marked in blue.
>
>     Link: https://openreview.net/pdf?id=N4NhVN30ph
>
> ====================================================
>
> **Off-policy methods are hard to train?**
>
> > The authors claim that off-policy algorithms are often more sample efficient than on-policy counterparts. However, PPO is still often the way to go when you face a new RL problems because of better stability and the need for less hyperparameter tuning. I am not fully convinced this approach won't have the same issue.
> >
>
> We agree with the reviewer's insightful observation. Training off-policy methods is indeed generally more challenging than on-policy methods, as off-policy approaches aim to maximize sample efficiency by reusing past experiences. This advantage, however, comes at the cost of more complex update rules, intricate model architectures, and increased hyperparameter tuning. Nonetheless, off-policy methods are advantageous in environments with costly interactions, significantly reducing the need for new samples. Our method enhances this advantage by generating structured, correlated trajectories through movement primitives and achieving efficient value estimation via a transformer critic and N-step return.
>
> **More explanation of importance sampling?**
>
> > I believe the explanations of why importance sampling is not necessary could be improved.
> >
>
> Thank you for pointing this out. The use of importance sampling in classical off-policy RL methods arises from the need to estimate the value $Q^{\pi_{new}}(s_t, a_t)$ of a single action $a_t$  under the current policy $\pi_{new}$ using the rewards generated by future actions $a_{t+1}, …a_{t+N}$ sampled from an old behaviour policy $\pi_{b}$. The Q-function is not directly aware of these future actions, as they are not part of the input; however, we rely on their outcomes (e.g., rewards) to compute the target for the Q-function under $\pi_{new}$. Because the probability of sampling these actions differs under $\pi_{new}$, importance sampling is necessary, as outlined in Section 3.1 in our paper and the RL book [1] (Chapter 7.3, n-step Off-policy Learning).
>
> In contrast, when predicting the value $Q(s_t, a_t, a_{t+1}, …, a_{t+N})$ of an action sequence in TOP-ERL, we directly provide these actions $a_t, a_{t+1}, …, a_{t+N}$ as input to the critic transformer. Here, these future actions are deterministic for the Q-function rather than being sampled from a policy, eliminating the need for importance sampling to address varying sampling probabilities.
>
> **Can transformer attend more than one state?**
>
> > It is not completely clear to me if the transformers can attend only the current observation s^k_0 or also the previous ones during training?
> >
>
> In TOP-ERL, we use only a single state $s^k_0$ in the transformer, as we assume the tasks follow an MDP structure, where the task and environment status can be fully represented by a single observed state. Consequently, it is unnecessary to input more than one state.
>
> However, if the task is partially observable, we believe our method could incorporate multiple observations in the transformer critic to infer the underlying system and environment state, potentially improve performance, as discussed in the future work section. Additionally, serveral works in the literature have highlighted the benefits of using transformers in POMDP and scenarios that need memories, such as the work in [2].
>
> **Role played by MP?**
>
> > I am wondering how important are the probabilistic MP in the performance, what if the policy was also a transformer?
> >
>
> MP approaches, as parameterized trajectory generators, have been widely used in fields such as robot learning and path planning [3-8]. The primary advantage of MP is its ability to generate smooth, temporally- and DoF-correlated trajectories, which has been shown to significantly enhance task performance, as evidenced by [6] (one of the baseline methods in our experiment). Additionally, MP functions as a macro-scale skill representer, predicting the entire action sequence (100-500 steps in our experiments) in a single pass.
>
> Directly using transformers as policies is indeed a growing research area, though it remains mostly limited to imitation learning and offline RL, where they are used to predict action chunks and regress behaviors from fixed datasets, typically spanning 3-20 steps.
>
> (To be continued ...)

---

> ### Author Response · Authors · 2024-11-18
> **Reply to reviewer iE4K (part 2)**
>
> (Continue from part 1...)
>
> In online RL, however, the use of action chunks remains largely unexplored. We attribute this to challenges such as complex, unstable update rules and the difficulty in maintaining consistency for generating long-horizon, structured trajectories during exploration. Previous works, such as [9], augment SAC by incorporating an RNN as the action generator. However, most tasks in this work only predict 3-step actions. In contrast, in our work, we predict 100-500 actions using MPs.
>
> **Positional encoding for similar states at two different time steps?**
> > I am also wondering if the positional encoding of two different states s^k_0 and s'^k_0 would always be the same (even if one is more in the future that the other one)?
> >
>
> Since we are working on episodic tasks with a finite horizon, the task state includes a unique timestamp, allowing the agent to differentiate the value of two similar states at different time steps. This design choice aligns with findings in the literature, such as [10]. Consequently, the positional encoding is solely responsible for distinguishing the order of the segment’s initial state and the subsequent actions within the input sequence.
>
> **Remove the Target net?**
>
> > Is a target network necessary at all?
> >
>
> We thank the reviewer for this interesting idea. In response, we conducted an additional ablation study by removing the target network from our model and observed a performance drop, as detailed in Section R.5 of the supplementary material. Target networks are essential for stabilizing training in classical off-policy methods. However, recent work [11] has introduced novel normalization techniques that reduce the need for target networks, which could potentially be integrated into our transformer-based critic. We leave this as an open topic for future research.
>
> ## References
>
> [1] Sutton, Richard S. "Reinforcement learning: An introduction." *A Bradford Book* (2018).
>
> [2] Ni, Tianwei, et al. "When do transformers shine in RL? decoupling memory from credit assignment." NeurIPS 2023.
>
> [3] Peters, J., Vijayakumar, S. and Schaal, S., 2005. Natural actor-critic. In *Machine Learning: ECML 2005*.
>
> [4] Kober J, Peters J. Policy search for motor primitives in robotics. NIPS 2008.
>
> [5] Otto F, Celik O, Zhou H, Ziesche H, Ngo VA, Neumann G. Deep black-box reinforcement learning with movement primitives. In Conference on Robot Learning 2022.
>
> [6] Li Ge, et al. “Open the Black Box: Step-based Policy Updates for Temporally-Correlated Episodic Reinforcement Learning.” ICLR 2024.
>
> [7] Kicki P, Tateo D, Liu P, Guenster J, Peters J, Walas K. Bridging the gap between learning-to-plan, motion primitives and safe reinforcement learning. In Conference on Robot Learning 2024.
>
> [8] Onur Celik, et al. "Acquiring Diverse Skills using Curriculum Reinforcement Learning with Mixture of Experts.", ICML 2024. **
>
> [9] Zhang, Haichao, Wei Xu, and Haonan Yu. "Generative Planning for Temporally Coordinated Exploration in Reinforcement Learning." ICLR 2022
>
> [10] Jackson, Matthew Thomas, et al. "Discovering Temporally-Aware Reinforcement Learning Algorithms." ICLR 2024.
>
> [11] Bhatt, Aditya, et al. "CrossQ: Batch Normalization in Deep Reinforcement Learning for Greater Sample Efficiency and Simplicity." ICLR 2024.
>
> ================================================
>
> If you have any further questions, please feel free to ask us.
>
> Thank you.
>
> TOP-ERL Authors

---

> > ### Author Response · Authors · 2024-11-25
> > **We look forward to your response.**
> >
> > Dear reviewer iE4K,
> >
> > As the rebuttal period comes to an end, we would like to confirm if our responses have sufficiently addressed your concerns. Thank you for your time and support.
> >
> > Best,
> >
> > TOP-ERL Authors

---

> > > ### Comment · Reviewer_iE4K · 2024-11-25
> > >
> > > Thank you for answer all the question and running the target network ablation in this short period, I do not have further one.

---

> > > > ### Author Response · Authors · 2024-11-25
> > > > **Thank you**
> > > >
> > > > Dear Reviewer,
> > > >
> > > > Thank you for taking the time to review our paper and for your approval of our work.
> > > >
> > > > Best wishes,
> > > >
> > > > TOP-ERL Authors.

---

### Official Review · Reviewer_bGrR · 2024-11-01

**Soundness:** 2
**Presentation:** 3
**Contribution:** 2
**Rating:** 8
**Confidence:** 3

**Summary:**

The paper proposes a new off-policy episodic reinforcement learning framework that, especially, improves sampling efficiency over traditional episodic-based and step-based baselines. A main novelty of the work is using a Transformer-based architecture to train a critic network that predicts value and Q-functions for a sequence of actions from an initial state. This critic network is then adapted and applied in an SAC-like algorithm for off-policy episodic reinforcement learning training.

**Strengths:**

1.	A critic network for action sequences using a traditional Transformer architecture is proposed to model N-step returns.
2.	A SAC-like algorithm is adapted for off-policy episodic reinforcement learning using the trained critic network.
3.	The proposed methodology is shown to improve sampling efficiency and also is shown to stabilize training over other baseline methods.

**Weaknesses:**

1.	In the current writing, it is hard to identify the novel parts that are proposed by the author and parts that are kept from other works.
2.	Section 4.4 seems very important to understand how to adapt SAC to episodic reinforcement learning but it seems not clear.
3.	Given the resemblance of the critic training to the prediction of rewards that can be used for more effective credit assignment compared to dense and sparse rewards, additional ablation studies or baselines could be discussed.

**Questions:**

Comments:

1.	In Eq. (8), the future return after N-step is modeled by the critic network during training? It is interesting that if this is the case, the training is stable.
2.	In the reviewer’s view, it is hard to grasp the need for the enforcement of the initial condition in section 4.3.1 as s_0^k is already a condition to generate the trajectory in the policy. Some additional intuition about how this is related to the division in segments, or because of the variable length used, or the sampling of trajectories would clarify this part to the reader.
3.	Another critical part is Section 4.4.  It is not clear to the reviewer how SAC is adapted to the new policy objective during training.
4.	In Fig.4, why for (c) and (d), the TOP-ERL training is stopped before the other baselines?
5.	The impact of choosing random segments seems to be crucial to performance. Any intuition regarding this result?
6.	Given the resemblance of the critic training to the prediction of rewards that can be used for more effective credit assignment compared to dense and sparse rewards, the reviewer thinks what is the relation between the proposed methodology and using other networks that predict future rewards for more effective credit assignment in performance for the robotic tasks in Section 5.

Minor Comments:

1.	Line 62: Typo “Transformer”
2.	Lines 124-130: The writing in these lines is very confusing. It could be rewritten more clearly.
3.	Title Section 4.5: Is it “summarize” the right word here?
4.	Line 425: Typo “empirical”
5.	Line 439: Typo “Noteably”
6.	Line 471: Typo “sprase”

---

> ### Author Response · Authors · 2024-11-18
> **Reply to reviewer bGrR**
>
> **Dear Reviewer bGrR,**
>
> We are grateful for the time and effort you have dedicated to reviewing our submission. In response to your valuable feedback, we have revised our paper and have addressed your concerns to the best of our knowledge, as detailed below.
>
> **Update Log [Nov. 18]**
>
> - Address the concerns of all reviewers.
> - Upload additional experiment and ablation study to Supplementary Material.
>
>     Link: https://openreview.net/attachment?id=N4NhVN30ph&name=supplementary_material
>
> - Revise the manuscript, with modifications marked in blue.
>
>     Link: https://openreview.net/pdf?id=N4NhVN30ph
>
> =======================================================
>
> **Revise to identify the novelty**
>
> > In the current writing, it is hard to identify the novel parts that are proposed by the author and parts that are kept from other works.
> >
>
> We apologize for any lack of clarity in our initial writing. We have revised the paper in Section 4, by explicitly distinguish the contribution made by our model (Section 4.2-4.4) and the techniques adopted from the literature (Section 4.1 and Section4.5).
>
> **More details for policy training in section 4.4?**
>
> > Section 4.4 seems very important to understand how to adapt SAC to episodic reinforcement learning but it seems not clear.
> >
>
> We apologize for the insufficient presentation in our initial submission. We have expanded this section with additional details and included it in the supplementary material Section R.2.
>
> **Discussion on the Relationship to Reward Prediction Methods?**
> > Given the resemblance of the critic training to the prediction of rewards that can be used for more effective credit assignment compared to dense and sparse rewards, additional ablation studies or baselines could be discussed.
> >
>
> We thank the reviewer for this insightful question. Our methodology extends SAC [1] by employing N-step TD error for critic training, which estimates the expected return for a given action sequence. Unlike approaches that explicitly predict future rewards, such as in model-based RL, our method avoids constructing a reward or transition model, which can introduce biases and instability, particularly in robotic tasks with complex dynamics. Instead, N-step TD error facilitates effective credit assignment by capturing the cumulative impact of actions over multiple steps.
>
> While model-based methods or reward prediction networks can enhance credit assignment, they require additional modeling complexity and may suffer from inaccuracies in the learned models. Our approach benefits from the simplicity and stability of TD-based updates, seamlessly integrating into the SAC framework. We consider exploring comparisons with model-based or reward prediction-based methods in robotic tasks to be an interesting direction for future work.
>
> **Future return after N-step?**
> > In Eq. (8), the future return after N-step is modeled by the critic network during training? It is interesting that if this is the case, the training is stable.
> >
>
> Yes, the future return after N steps is predicted by the target critic network, which is a delayed version of the critic network being optimized. Additionally, we use a V-function $V_{\phi_{tar}}$ instead of a Q-function $Q_{\phi_{tar}}$ to bootstrap the future return after N-steps. This approach is computationally cheaper, as it requires no action sequence input, while still achieving comparably good performance.
>
> Notably, using a V-function to bootstrap the future return aligns with the original SAC approach, as shown in Eq. (2) of [1], where a V-function is used to bootstrap the one-step future return. However, in this early SAC version, the V- and Q-functions were trained independently with separate neural networks. Due to the complexity of training two networks, this version is updated to the version using a Q-function for bootstraping. In contrast, our method employs a shared transformer architecture and parameters for both the V- and Q-functions and do not explicitly train two networks.
>
> **More discussion on initial condition enforcement?**
>
> > In the reviewer’s view, it is hard to grasp the need for the enforcement of the initial condition in section 4.3.1 as s_0^k is already a condition to generate the trajectory in the policy. Some additional intuition about how this is related to the division in segments, or because of the variable length used, or the sampling of trajectories would clarify this part to the reader.
> >
>
> We apologize for the unclear explanation. In ERL, the policy typically predicts an entire action trajectory based on the initial task description, i.e. at time $t=0$. Our method, TOP-ERL, is an off-policy approach, meaning the current policy is updated using data collected from a behavior policy that interacted with the environment, which we refer to as the old policy.
>
> (To be continued...)

---

> > ### Author Response · Authors · 2024-11-18
> > **Reply to reviewer bGrR (part 2)**
> >
> > (Continue from part one...)
> >
> > We highlight the difference between the trajectories predicted by the two policies in the newly added Figure R1 in the supplementary material Section R1. The new action trajectory (dashed red) predicted by the current policy aligns with the old trajectory (solid black) at $t=0$, but gradually deviates as time progresses. In the critic update step, as described in Eq. (7), we input the old state $s_0^k$ from the old trajectory and the new actions from the new trajectory into the transformer. However, these inputs mismatch. To address this, we leverage the dynamic system property of our trajectory generator, ProDMP[2], to enforce each new action segment to start from their correponding old state. More detailed explanation can be found in Section R1.
> >
> > **The training of the Box Pushing tasks are concluded at 14M?**
> > > In Fig.4, why for (c) and (d), the TOP-ERL training is stopped before the other baselines?
> > >
> > We concluded the training of our model at 14M, as it significantly outperformed the baseline methods in both performance and sample efficiency. However, we have included updated plots of these training curves up to 40M in the supplementary Section R4.
> >
> > **Intuition regarding the randm segment length?**
> > > The impact of choosing random segments seems to be crucial to performance. Any intuition regarding this result?
> > >
> > In addition to allowing the critic transformer to capture longer action correlations, we believe the critic training also benefits from incorporating a wide range of action sequence lengths. For example, in Eq. (7), the expectation of the Q-value over L actions is used as the target for the V-function's prediction. By varying L across update iterations, the V-function is trained on Q-values derived from different numbers of actions, which implicitly helps regularize the critic network training. Additionally, using random lengths eliminates the need to treat segment length as a hyperparameter.
> >
> > **Minor comments and typos**
> >
> > We appreciate the reviewer's attention to these writing details and have updated our text accordingly.
> >
> >
> > ## References
> >
> > [1] Haarnoja, Tuomas, et al. "Soft actor-critic: Off-policy maximum entropy deep reinforcement learning with a stochastic actor." ICML 2018.
> >
> > [2] Li, Ge, et al. "Prodmp: A unified perspective on dynamic and probabilistic movement primitives." *IEEE Robotics and Automation Letters* 8.4 (2023): 2325-2332.
> >
> > ================================================
> >
> > If you have any further questions, please feel free to ask us.
> >
> > Thank you.
> >
> > TOP-ERL Authors

---

> > > ### Comment · Reviewer_bGrR · 2024-11-22
> > > **Rebuttal Feedback**
> > >
> > > Thank you for addressing and clarifying the reviewer's concerns and comments.
> > >
> > > I have increased my score accordingly.

---

> > > > ### Author Response · Authors · 2024-11-22
> > > > **Thank you for updating your recommendation**
> > > >
> > > > Dear reviewer,
> > > >
> > > > Thank you for your reply! We sincerely appreciate your approval of our work.
> > > >
> > > > Best regards,
> > > >
> > > > TOP-ERL Authors

---

### Official Review · Reviewer_gDNi · 2024-11-03

**Soundness:** 2
**Presentation:** 3
**Contribution:** 2
**Rating:** 6
**Confidence:** 3

**Summary:**

This paper presents TOP-ERL, Transformer-based Off-Policy Episodic RL, which leverages the Transformer as a critic to predict the value of action sequences, splits it into smaller segments and inputs them into the Transformer for value prediction, with an off-policy update scheme.

**Strengths:**

This paper is easy to follow. Given that research on episodic RL is relatively limited, TOP-ERL shows its strengths in sample efficiency and overall performance.

**Weaknesses:**

1. The code repository is incomplete, lacking the environment configuration files and key training files. Additionally, several important functions, such as ValueFunction, have not been implemented.

2. Figure 4(a) only presents the average results for the 50 tasks, without showing the individual results for each task. While I understand that the main text has space constraints, I strongly recommend including the results for all 50 tasks in the appendix.

3. The methodology seems to lack novelty, as important components follow the design of previous episodic RL methods. I did not observe other core contributions aside from the reported strong performance. In the experimental section, it would be beneficial to see experiments and analyses related to the gaussian policy and movement primitives module.

**Questions:**

1. The paper claims to be "the first off-policy episodic reinforcement learning algorithm," but this seems to be an overclaim. To my knowledge, there are already several related works, such as:
Liang D, Zhang Y, Liu Y. "Episodic Reinforcement Learning with Expanded State-reward Space." arXiv preprint arXiv:2401.10516, 2024.

2. Regarding the statement about "enforcing action initial conditions," can this approach effectively address the mismatch between the state and the corresponding action sequence? In addition to the ODE explanation, could you design experiments to validate this claim?

3. Why does using random segments yield better results than most fixed segments? Could you provide a more intuitive explanation? The correlation presented in Appendix D does not seem to clarify this adequately.

---

> ### Author Response · Authors · 2024-11-18
> **Reply to reviewer gDNi**
>
> Dear reivewer gDNi,
>
> Thank you for your valuable and insightful feedback on our work. We have addressed your concerns to the best of our knowledge.
>
>
> **Update Log [Nov. 18]**
>
> - Address the concerns of all reviewers.
> - Upload additional experiment and ablation study to Supplementary Material.
>
>     Link: https://openreview.net/attachment?id=N4NhVN30ph&name=supplementary_material
>
> - Revise the manuscript, with modifications marked in blue.
>
>     Link: https://openreview.net/pdf?id=N4NhVN30ph
>
> =====================================================
>
> **First off-policy ERL paper?**
>
> > The paper claims to be "the first off-policy episodic reinforcement learning algorithm," but this seems to be an overclaim. To my knowledge, there are already several related works, such as: Liang D, Zhang Y, Liu Y. "Episodic Reinforcement Learning with Expanded State-reward Space." arXiv preprint arXiv:2401.10516, 2024.
> >
>
> We thank the reviewer for pointing this out. We believe the question arises from the similarities between two distinct concepts: “episodic reinforcement learning” (ERL) and “episodic control-based reinforcement learning” (ECRL).
>
> In our work, we refer to episodic reinforcement learning as defined in foundational RL works such as Episodic REINFORCE [1], Episodic Natural Actor Critic [2], and Episodic Policy Learning [3], which has been further developed in recent studies [4, 5, 6, 7, 8]. This line of research frames the RL problem as optimizing controller parameters to maximize cumulative return across each episode. To the best of our knowledge, TOP-ERL is the first off-policy algorithm developed within this framework.
>
> In contrast, the work by Liang et al. [9], referenced by the reviewer, aligns with a different line of research rooted in Episodic Control (EC) [10, 11]. ECRL typically incorporates a specialized episodic memory to regularize Q-function learning by reusing trajectories with similar state-action pairs (as discussed in Section 2.3 of [9]).
>
> We plan to include a discussion in a future version of our paper to clarify the distinctions between these two concepts and address potential misunderstandings.
>
> **Concerns of novelty**
>
> > The methodology seems to lack novelty, as important components follow the design of previous episodic RL methods. I did not observe other core contributions aside from the reported strong performance. In the experimental section, it would be beneficial to see experiments and analyses related to the gaussian policy and movement primitives module.
> >
> Thank you for your feedback. We have revised several parts of Section 4 to explicitly differentiate the contributions of our model from the techniques adopted from the literature.
>
> The core contributions of our approach are summarized as follows: (1) we introduce a method integrating a Transformer as the critic within model-free, online RL framework, (2) we propose using N-step return as the target for the Transformer critic, and (3) we present TOP-ERL as the first Off-Policy ERL algorithm. Each of these contributions is novel compared to previous ERL methods.
>
> The benefits of using Movement Primitives instead of standard Gaussian policy have already been well discussed in previous ERL works, including generating smoother trajectories [4] and capturing the temporal correlations between actions [5]. We briefly discussed these features in the related works (Section 2).
>
> **Incomplete Repository?**
> > The code repository is incomplete, lacking the environment configuration files and key training files. Additionally, several important functions, such as ValueFunction, have not been implemented.
> >
> We apologize for the missing configs. These configs included personal working directories and HPC usernames, so we initially excluded them to comply with double-blind review rules. We have now cleaned and uploaded them (four configs with the keyword “hpc”) to the anonymous GitHub repository.
>
> The implementations of the value functions were included in the initial code base at https://github.com/toperliclr2025/TOP_ERL/tree/main, with locations listed as below:
>
> - The transformer backbone underlying the critic is implemented in *util_nanogpt.py (*https://github.com/toperliclr2025/TOP_ERL/blob/main/mprl/util/util_nanogpt.py*),* adapted from the open-source *NanoGPT* library (https://github.com/karpathy/nanoGPT)
> - The critic class, including components for various configurations such as double Q-networks and target networks, is in *seq_critic.py* (https://github.com/toperliclr2025/TOP_ERL/blob/main/mprl/rl/critic/seq_critic.py).
> - The RL agent, including the N-step return (Eq. 8 in the paper), is in *seq_agent.py* (https://github.com/toperliclr2025/TOP_ERL/blob/main/mprl/rl/agent/seq_agent.py)*,* with the N-step return implementation on line 304. The variant using the Q-function as the future return is on line 585.
>
> (To be contined...)

---

> > ### Author Response · Authors · 2024-11-18
> > **Reply to reviewer gDNi (part two)**
> >
> > (Continue from part one...)
> >
> > For environments and their configurations, we used several open-source repositories, including *Metaworld* (https://github.com/Farama-Foundation/Metaworld) and *FancyGym* (https://pypi.org/project/fancy-gym/). For the trajectory generator, we used the open-source *MP_PyTorch* library (https://pypi.org/project/mp-pytorch/).
> >
> >
> > **More discussion about initial condition enforcement?**
> > > Regarding the statement about "enforcing action initial conditions," can this approach effectively address the mismatch between the state and the corresponding action sequence? In addition to the ODE explanation, could you design experiments to validate this claim?
> > >
> >
> > We include additional discussions and provide a figure of our experiment in Section R.1 of the supplementary material, demonstrating this enforcement in the box-pushing task.
> >
> > **More intuition of random segment length?**
> > > Why does using random segments yield better results than most fixed segments? Could you provide a more intuitive explanation? The correlation presented in Appendix D does not seem to clarify this adequately.
> > >
> >
> > In addition to enabling the critic transformer to capture longer action correlations, we believe the critic training also benefits from using a wide range of action sequence lengths. For instance, in Eq. 7, we use the expectation of the Q-value over L actions as the target for the V-function's prediction. When L varies across different update iterations, the V-function is trained on Q-values derived from different amounts of actions, which implicitly regularizes the training of the critic network. Furthermore, using a random length removes the need to tune it as a hyperparameter.
> >
> > **Each task in Metaworld?**
> > > Figure 4(a) only presents the average results for the 50 tasks, without showing the individual results for each task. While I understand that the main text has space constraints, I strongly recommend including the results for all 50 tasks in the appendix.
> > >
> >
> > Thank you for pointing this out. We now have included all the 50 subtasks in the supplementary material, in section R.7.
> >
> > **References**
> >
> > [1] R. J. Williams. Simple statistical gradient-following algorithms for connectionist reinforcement learning. Machine Learning, 8:229–256, 1992.
> >
> > [2] Peters, J. et al. Natural actor-critic. In *Machine Learning: ECML 2005.*
> >
> > [3] Kober J, Peters J. Policy search for motor primitives in robotics. NIPS 2008.
> >
> > [4] Otto, Fabian et al. Deep black-box reinforcement learning with movement primitives. In Conference on Robot Learning 2022.
> >
> > [5] Li, Ge et al. Open the Black Box: Step-based Policy Updates for Temporally-Correlated Episodic Reinforcement Learning. ICLR 2024.
> >
> > [6] Onur Celik, et al. "Acquiring Diverse Skills using Curriculum Reinforcement Learning with Mixture of Experts.", ICML 2024. **
> >
> > [7] Kicki, Piotr et al. Bridging the gap between learning-to-plan, motion primitives and safe reinforcement learning. In Conference on Robot Learning 2024.
> >
> > [8] Hüttenrauch, Maximilian et al. Robust Black-Box Optimization for Stochastic Search and Episodic Reinforcement Learning. JMLR 2024.
> >
> > [9] Liang D, Zhang Y, Liu Y. Episodic Reinforcement Learning with Expanded State-reward Space. AAMAS 2024
> >
> > [10] Blundell C, Uria B, Pritzel A, Li Y, Ruderman A, Leibo JZ, Rae J, Wierstra D, Hassabis D. Model-free episodic control. arXiv preprint arXiv:1606.04460. 2016 Jun 14.
> >
> > [11] Pritzel A, Uria B, Srinivasan S, Badia AP, Vinyals O, Hassabis D, Wierstra D, Blundell C. Neural episodic control. ICML 2017.
> >
> > ================================================
> >
> > If you have any further questions, please feel free to ask us.
> >
> > Thank you.
> >
> > TOP-ERL Authors

---

> > > ### Comment · Reviewer_gDNi · 2024-11-27
> > >
> > > Thank you for your detailed response to my review. However, I still have some concerns.
> > >
> > > I reviewed the authors’ commit history on their GitHub repository (https://github.com/toperliclr2025/TOP_ERL/tree/main). While a config file was added last week, the main `train.py` file is still missing, and there is no essential code documentation. Additionally, the repository lacks necessary environment configuration files such as `requirements.txt` or `setup.py`. I remain skeptical about the reproducibility of the code and recommend the authors polish the code repository thoroughly.
> > >
> > > I understand that integrating the Transformer as a critic is a complex and ingenious design. However, the experiments lack direct validation of the critic transformer's effectiveness, such as reporting metrics like critic error.
> > >
> > > Thank you for providing detailed performance results. However, it is noticeable that TOP-ERL does not demonstrate a prominent success rate and superior sample efficiency in many environments.
> > >
> > > So I am inclined to keep my original score.

---

> > > > ### Author Response · Authors · 2024-11-29
> > > > **Reply to remaining concerns (part 1)**
> > > >
> > > > Dear reviewer,
> > > >
> > > > Thank you for your reply and we address your concerns as follows:
> > > >
> > > > ### **Enhancing Reproducibility?**
> > > >
> > > > > I reviewed the authors’ commit history on their GitHub repository (https://github.com/toperliclr2025/TOP_ERL/tree/main). While a config file was added last week, the main `train.py` file is still missing, and there is no essential code documentation. Additionally, the repository lacks necessary environment configuration files such as `requirements.txt` or `setup.py`. I remain skeptical about the reproducibility of the code and recommend the authors polish the code repository thoroughly.
> > > > >
> > > >
> > > > We completely agree with your point that better reproducibility increases researchers’ confidence in our work and benefits the community. To reply your questions:
> > > >
> > > > - On Nov 17, we have added **three** (instead of only one) HPC config files to GitHub. Their commit numbers are listed below:
> > > >
> > > >     - Metaworld: a8dfb6606cc6c2e9d42cc0b241163340ad56eb53
> > > >
> > > >     - Hopper Jump: 13a10455cbcd0d401dce01c14aede026beb21233
> > > >
> > > >     - Box push: f9eb3be5a4e3399f3233300ab4a92667df1ca2f2
> > > >
> > > > - The `train.py` file was already included in our original code base under the name `seq_mp_multiprocessing.py` (https://github.com/toperliclr2025/TOP_ERL/blob/main/mprl/seq_mp_exp_multiprocessing.py). This file instantiate each component of our model and uses multiprocessing techniques to run environment rollouts and RL model training in parallel processes, which speeds up the learning process.
> > > > - To enable one-click environment setup, we have added a Mamba (Conda) environment installation file, `conda_env.sh` (https://github.com/toperliclr2025/TOP_ERL/blob/main/conda_env.sh). We also updated our README file with detailed installation guidelines.
> > > > - Additionally, we recorded a **tutorial video** (https://www.youtube.com/watch?v=y-d1E0qkZFM&t=616s) to provide a step-by-step guide for environment installation and experiment execution. The tutorial demonstrates the procedure on a fresh Linux PC with GitHub access. The entire process, from installing Mamba-forge to running experiments, takes **only 10 minutes.** The video details each step as follows:
> > > >     - 0:00 Introduce to our PC setup
> > > >     - 0:22 Install Mamba (a release of conda with faster speed)
> > > >     - 1:48 Clone TOP-ERL repository from GitHub
> > > >     - 2:22 Install all dependencies in one click
> > > >     - 6:18 Log in to WandB in a web browser
> > > >     - 8:30 Log in to WandB on the local PC
> > > >     - 8:43 Adapt the user’s WandB entity in TOP-ERL's config file
> > > >     - 9:36 Run the experiment, and monitor it in WandB
> > > >
> > > >     To comply with double-blind review principles, all accounts shown in the video are anonymous.
> > > >
> > > > - We included an additional statement in this forum to highlight the reproducibility of our work.
> > > >
> > > > ### **Effectiveness of the critic Transformer?**
> > > > > I understand that integrating the Transformer as a critic is a complex and ingenious design. However, the experiments lack direct validation of the critic transformer's effectiveness, such as reporting metrics like critic error.
> > > > >
> > > >
> > > > In the supplementary material submitted last week with the rebuttal, we included a comparison between the transformer critic conditioned on action sequences and its ablated version conditioned on single actions, which we refer to as the Degenerated Transformer. We reported its performance in the box-pushing dense reward setting in Figure R4, where it performed significantly worse than TOP-ERL.
> > > >
> > > > To further evaluate the effectiveness of the transformer critic, we adopted the analysis approach outlined in Clipped Double Q-learning [1]. Figure R5 compares the average critic predictions to the average Monte Carlo returns, while Figure R6 illustrates the difference between these values, referred to as the critic bias.
> > > >
> > > > The results demonstrate that the transformer critic in TOP-ERL predicts values with less bias, whereas the ablated model shows clear overestimation in its predictions. We infer that the use of action sequences implicitly regularizes the training of the value function. Furthermore, as off-policy RL methods rely on the critic to guide policy updates, the transformer-based critic enables decision-making over entire action sequences rather than individual per-step actions. This capability significantly enhances training efficiency and improves overall task performance.
> > > >
> > > > ### Reference
> > > >
> > > > [1] Fujimoto, Scott, Herke Hoof, and David Meger. "Addressing function approximation error in actor-critic methods." ICML, 2018.
> > > >
> > > > (To be continued...)

---

> > > > > ### Author Response · Authors · 2024-11-29
> > > > > **Reply to remaining concerns (part 2)**
> > > > >
> > > > > (continue from above...)
> > > > >
> > > > > ### **Why is TOP-ERL not outperforming all 50 tasks in Metaworld?**
> > > > > > Thank you for providing detailed performance results. However, it is noticeable that TOP-ERL does not demonstrate a prominent success rate and superior sample efficiency in many environments.
> > > > > >
> > > > >
> > > > > Metaworld consists of 50 distinct tasks with varying types and difficulties. We did not independently optimize TOP-ERL for each task but instead used the same hyperparameters across all of them. Similar to TOP-ERL, all reported algorithms show variable performance across different tasks.
> > > > >
> > > > > However, we want to highlight that **TOP-ERL is either the best or comparably outperforming in approximately 45 of these tasks, which explains its overall superior performance in Figure 4(a).**
> > > > >
> > > > > ==============================================
> > > > >
> > > > > If you have any further questions, please feel free to reach out to us. If our efforts have addressed your concerns, including the previous ones, we kindly ask you to consider increasing your recommendation. Your approval is very important to us.
> > > > >
> > > > > Thank you.
> > > > >
> > > > > TOP-ERL Authors

---

> > > > > > ### Author Response · Authors · 2024-12-02
> > > > > > **We Look Forward to Your Reconsideration**
> > > > > >
> > > > > > Dear Reviewer,
> > > > > >
> > > > > > As the rebuttal period is nearing its conclusion, we are eager to know if our recent and previous responses have sufficiently addressed your insightful concerns.
> > > > > >
> > > > > > Since your recommendation score remains at **5**—slightly below the acceptance threshold—we would be deeply grateful if you could consider increasing it to reflect the efforts we have made to address your feedback during the rebuttal phase. Your support and endorsement are essential for our work to be evaluated as a high-quality contribution to ICLR.
> > > > > >
> > > > > > Thank you for your time, thoughtful engagement, and valuable feedback throughout this process.
> > > > > >
> > > > > >
> > > > > > Best regards,
> > > > > >
> > > > > > TOP-ERL Authors

---

> ### Author Response · Authors · 2024-11-25
> **We look forward to your response.**
>
> Dear Reviewer gDNi,
>
> As the rebuttal period comes to an end, we would like to know if our responses have adequately addressed your concerns. Your feedback will help us make a more informed decision about our submission. Thank you in advance for your time and input.
>
> Best,
>
> TOP-ERL Authors

---

### Author Response · Authors · 2024-11-18
**Update Log**

We thank all reviewers for their valuable feedback and insightful suggestions, which have significantly improved the quality of our paper. Below, we provide an update log summarizing the changes made to the paper.


**Update Log [Nov. 18, 2024]**

- Reply to all reviewers regarding their major concerns
- We have updated the supplementary material with additional discussions and ablation studies, using indices starting with “R” to specify the new content added during the rebuttal phase.
    - Section R.1: Detailed discussion of initial condition enforcement
    - Section R.2: Detailed discussion of policy training
    - Section R.3: Technical details regarding the TRPL for stable policy training
    - Section R.4: Longer training steps for box pushing tasks
    - Section R.5: Ablation to removing the target critic net in our method
    - Section R.6: Two ablation cases to demonstrate the effectiveness of the critic transformer
    - Section R.7: Individual Performance Report for Metaworld 50 Tasks

**Update Log [Nov. 27, 2024]**
- Section R.6: Adding two additional figures to analyze the impact of transformer critic on value predictions


**Update Log [Nov. 29, 2024]**
- We added a tutorial [video](https://www.youtube.com/watch?v=y-d1E0qkZFM&t=616s&ab_channel=top_erl_iclr25) to guide the installation and running experiment of our work.


**Update to Paper [Nov. 18, 2024]**

- Section 2, revise the discussion of several offline RL approaches.
- Section 4, explicitly differentiate the contributions of our model from the techniques adopted from the literature.
- Section 5, continue training TOP-ERL in box pushing tasks till 40M

**Update to Paper [Nov. 26, 2024]**

- Consolidated several discussions of technical details from the supplementary material into the main manuscript's appendix.


Link to supplementary: https://openreview.net/attachment?id=N4NhVN30ph&name=supplementary_material

Link to updated paper: https://openreview.net/pdf?id=N4NhVN30ph

==============================================

Thank you,

TOP-ERL Authors

---

### Author Response · Authors · 2024-11-29
**Reproducibility Statement**

# Reproducibility Statement

Dear reviewers and readers,

We acknowledge concerns regarding the installation of our code base and the reproduction of our experiments. We have taken the following steps to address these concerns:

- In our initial submission, we provided an anonymous GitHub repository containing all implementations of our methods, including the critic transformer, the RL agent, and the config file for running experiments in a local PC.

- We further updated our code base by offering configs for slurm-based HPC systems.

- To enable one-click environment setup, an environment configuration file `conda_env.sh` (https://github.com/toperliclr2025/TOP_ERL/blob/main/conda_env.sh) has been added to streamline the setup process.

- We recorded a **tutorial video** (https://www.youtube.com/watch?v=y-d1E0qkZFM&t=616s&ab_channel=top_erl_iclr25) to guide users step-by-step through the installation and experiment procedures. Using a fresh PC with GitHub access, the entire process—from installing mamba-forge (a fast conda distribution) to starting the experiment—takes  **only 10 min**. To adhere to double-blind review principles, all accounts shown in the video are anonymized.

We believe these efforts address potential concerns regarding the reproducibility of our work.

Best regards,

TOP-ERL Authors

---

### Meta-Review · Area_Chair_RS8a · 2024-12-21

**Metareview:**

The paper proposes a method for off-policy RL that segments trajectories and then feeds them to a transformer critic, which is updated using an off-policy update. Empirical results on robot learning environments (Meta-World) show that the proposed method outperforms prior methods. The paper also detailed ablations.

Reviewers appreciated the clear writing and motivation of the paper, alongside the strong empirical performance on a large-scale RL benchmark. They also appreciated the novelty in how the transformer was used (modeling V and Q with the same architecture).

In terms of experiments, reviewers noted that the proposed method seems to be heavily reliant on some techniques (trust region constraints, layer normalization), raising questions about whether the performance of the proposed method is _solely_ explained by these decisions (rather than the use of the transformer). One reviewer requested an ablation of the movement primitive module, and another raised questions about the completeness of the released code. In terms of writing, reviewers suggested that the paper more clearly delineate novel contributions.

Overall, the reviewers unanimously voted to accept the paper. This seems like a strong paper (and even more so after the rebuttal) so I recommend acceptance.

**Additional Comments On Reviewer Discussion:**

The authors addressed concerns about reproducibility by releasing anonymous code and very detailed usage instructions. The authors also made several revisions to the paper, included new ablation experiments, and addressed several questions raised by the reviewers. Several reviewers increased their scores during the discussion.

---

### Decision · Program_Chairs · 2025-01-22

Accept (Spotlight)